# Multi-omics analysis defines highly refractory RAS burdened immature subgroup of infant acute lymphoblastic leukemia

Tomoya Isobe [1,2,3,33], Masatoshi Takagi [4,33] ✉, Aiko Sato-Otsubo[1,33], Akira Nishimura [4], Genta Nagae [5], Chika Yamagishi[4], Moe Tamura[2], Yosuke Tanaka [6], Shuhei Asada [6,7], Reina Takeda [6], Akiho Tsuchiya[6], Xiaonan Wang[3], Kenichi Yoshida [8], Yasuhito Nannya[8], Hiroo Ueno [8,9], Ryo Akazawa [9], Itaru Kato [9], Takashi Mikami [9], Kentaro Watanabe[1], Masahiro Sekiguchi [1], Masafumi Seki [1], Shunsuke Kimura[1,10], Mitsuteru Hiwatari[1,11], Motohiro Kato[1], Shiro Fukuda [5], Kenji Tatsuno [5], Shuichi Tsutsumi[5], Akinori Kanai [12], Toshiya Inaba[13], Yusuke Shiozawa [8], Yuichi Shiraishi[14], Kenichi Chiba[14], Hiroko Tanaka [15], Rishi S. Kotecha [16,17,18], Mark N. Cruickshank [19], Fumihiko Ishikawa [20], Tomohiro Morio [4], Mariko Eguchi [21], Takao Deguchi [22,23], Nobutaka Kiyokawa [24], Yuki Arakawa[25], Katsuyoshi Koh[25], Yuki Aoki[26], Takashi Ishihara [27], Daisuke Tomizawa [28], Takako Miyamura [29], Eiichi Ishii[21], Shuki Mizutani[4], Nicola K. Wilson [3], Berthold Göttgens [3], Satoru Miyano[15], Toshio Kitamura [6], Susumu Goyama [2], Akihiko Yokoyama[30], Hiroyuki Aburatani [5], Seishi Ogawa [8,31,32] & Junko Takita [1,9] ✉

*KMT2A*-rearranged infant acute lymphoblastic leukemia (ALL) represents the most refractory type of childhood leukemia. To uncover the molecular heterogeneity of this disease, we perform RNA sequencing, methylation array analysis, whole exome and targeted deep sequencing on 84 infants with *KMT2A*-rearranged leukemia. Our multi-omics clustering followed by single-sample and single-cell inference of hematopoietic differentiation establishes five robust integrative clusters (ICs) with different master transcription factors, fusion partners and corresponding stages of B-lymphopoietic and early hemato-endothelial development: IRX-type differentiated (IC1), IRX-type undifferentiated (IC2), HOXA-type MLLT1 (IC3), HOXA-type MLLT3 (IC4), and HOXA-type AFF1 (IC5). Importantly, our deep mutational analysis reveals that the number of RAS pathway mutations predicts prognosis and that the most refractory subgroup of IC2 possesses 100% frequency and the heaviest burden of RAS pathway mutations. Our findings highlight the previously underappreciated intra- and inter-patient heterogeneity of *KMT2A*-rearranged infant ALL and provide a rationale for the future development of genomics-guided risk stratification and individualized therapy.

Acute lymphoblastic leukemia (ALL) is the most common pediatric malignancy, in which up to 5% of cases are diagnosed during infancy (<12 months of age) and are biologically and clinically distinct from ALL in older children[1,2]. Infant ALL is characterized by a high frequency of *KMT2A* (also known as *MLL*) gene rearrangement (*KMT2A*-r) which is found in ~75% of cases and is associated with CD10-negative immature B-cell precursor phenotype and an extremely poor prognosis[3-5]. Although our most recent nationwide clinical trial, MLL-10, has significantly improved the survival of infants with *KMT2A*-r ALL[4], the event-free survival (EFS) rate is still below 40% worldwide[3,5].

The aggressive nature of the disease is reflected in its short latency between the occurrence of *KMT2A* rearrangement in utero[6,7] and overt disease progression in infancy. This short window period renders the infant ALL genome stable and mostly unaltered at diagnosis[8]. Among the few genetic events, frequent *RAS* mutations, as well as *FLT3* mutations and/or overexpression, have long been recognized[8-11]; nevertheless, translational applicability of these aberrations as risk factors and/or therapeutic targets for infant ALL is still under debate[5,8,12,13]. Therefore, in the treatment of infants with ALL, the positivity of *KMT2A* rearrangement remains the only established molecular risk factor[3,4], and no further molecular stratification is accepted worldwide, necessitating a better understanding of molecular heterogeneity within *KMT2A*-r infant ALL.

The recent development of machine learning methods for integrative subclass discovery from multi-omics data[14] has demonstrated excellent performance in discovering more complex and clinically relevant cancer subtypes than can be achieved by single-omics clustering[15-17]. Because *KMT2A*-r leukemia shares a common molecular signature[9] and therefore tends to cluster together against other subtypes of ALL[18,19], a multi-omics approach targeting only *KMT2A*-r cases would be the optimal design for uncovering the true heterogeneity within *KMT2A*-r infant ALL, although the rarity of the disease has hampered this approach to date.

In this study, we employ diagnostic samples of 84 infants with *KMT2A*-r leukemia, including B-ALL (*n* = 82) and B/myeloid mixed phenotype acute leukemia (B/M MPAL; *n* = 2), and perform omics-integrative clustering, followed by extensive subgroup characterization involving gene expression, DNA methylation, mutations, and copy number alterations. This multi-omics approach illustrates a truly unbiased molecular classification of *KMT2A*-r infant ALL and provides clues for genomics-driven diagnostics and potential therapeutic strategies.

## Results

### Transcriptome single-omics analysis validated the IRX and HOXA subtypes of *KMT2A*-r infant ALL

First, we performed RNA sequencing on diagnostic samples from 61 infants with *KMT2A*-r B-ALL (*n* = 59) or B/M MPAL (*n* = 2; Supplementary Data 1 and 2) as a discovery cohort. In total, 111 fusion transcripts were identified and experimentally validated (Supplementary Data 3), of which 95% were *KMT2A*-related fusions. In 25 of 61 cases (41%), alternative splicing of *KMT2A* fusions skipping the exon 11 of *KMT2A* was detected. Reciprocal *KMT2A* fusions (i.e., *X-KMT2A*) were observed in 17 cases (28%), mostly in *KMT2A-AFF1* cases (15 of 17 cases). In addition to *KMT2A* fusions, three cases (5%) harbored 5 *KMT2A*-unrelated fusions, two of which involved *AFF1* gene: *AFF1-SPATA13* and *AFF1-HSPA8* (Supplementary Data 3).

RNA sequencing-based gene expression clustering revealed two stable subgroups in *KMT2A*-r infant leukemia, marked by significant differential expression of *IRX* and *HOXA* transcription factors, as previously reported[13,20] (IRX subtype and HOXA subtype; Supplementary Fig. 1a, b). Network analysis further identified *IRX1* and *HOXA9* as the two top master regulators controlling a large number of downstream genes in each subtype (Supplementary Fig. 1c, d). Contrary to one previous report[12], expression of the reciprocal *AFF1-KMT2A* transcript

was not associated with the IRX/HOXA expression pattern (Supplementary Fig. 2a), whereas the *KMT2A* gene breakpoint was significantly associated with the IRX/HOXA pattern in *KMT2A-AFF1* cases (Supplementary Fig. 2b, c). Of note, while the IRX subtype has only been reported in *KMT2A-AFF1*-positive infants[12,13,20], we identified three *KMT2A-MLLT1*, one *KMT2A-EPS15* and one *KMT2A-AFF2* infants clustered within the IRX subtype (Supplementary Fig. 1b). A trend toward the previously reported poor prognosis of IRX subtype[13,20] was also observed among *KMT2A-AFF1* cases, whereas no survival differences were observed when all fusion partners were included (Supplementary Fig. 1e, f).

### Dual-omics consensus molecular subtypes of *KMT2A*-r infant ALL

Although gene expression-based clustering was able to validate the IRX/HOXA axis as a substructure of *KMT2A*-r infant ALL, in order to reveal more refined disease subgroups, we further analyzed the same 61 infant leukemia samples with DNA methylation array. As with transcriptome-based clustering, DNA methylation-based single-omics clustering failed to identify more than two stable clusters and divided the patients into *KMT2A-AFF1*-positive infants versus the others (Supplementary Fig. 3). However, since the two omics layers captured different key features for clustering the patients, we next sought to reclassify them by considering both transcriptome and DNA methylome. Except for clustering into two subgroups (*k* = 2; IRX vs. HOXA subtypes), similarity network fusion (SNF)-based dual-omics clustering[21] identified five subgroups as the most stable with respect to the stability metrics evaluated (Integrative Cluster 1–5 (IC1-5); Fig. 1a and Supplementary Fig. 4). Thereby, the IRX subtype was divided into IC1 and IC2, whereas the HOXA subtype was split into three ICs, which significantly correlated with fusion partners: *MLLT1* (IC3; Fisher's exact $P = 3.7 \times 10^{-8}$), *MLLT3* (IC4; Fisher's exact $P = 5.6 \times 10^{-6}$), and *AFF1* (IC5; Fisher's exact $P = 5.7 \times 10^{-6}$), respectively. Other patient characteristics, including known clinical prognostic factors such as age, did not correlate significantly with the IC assignment (Supplementary Table 1).

Importantly, these ICs were associated with clinical outcomes, wherein infants in IC2 exhibited the shortest EFS compared with IC1, IC4, or IC5 infants (log-rank $P = 1.5 \times 10^{-3}$, 0.017 and 0.042, respectively; Fig. 1b). Overall survival (OS) showed a similar trend, with IC2 infants showing the worst OS, although statistical significance was reached only between IC2 and IC1, the two ICs of the IRX subtype (log-rank $P = 0.015$; Fig. 1c). In a multivariate Cox regression model with clinical prognostic factors (Fig. 1d), ICs were significantly associated with EFS, independent of other prognostic parameters (with IC2 as the reference group; $P = 0.040$ and 0.046 for IC1 and IC5, respectively).

To validate these clustering results, we additionally performed DNA methylation array analysis in an independent cohort of 23 infants with *KMT2A*-r B-ALL (Supplementary Fig. 5a). Using a *k*-nearest neighbors (KNN) classifier modeled over the discovery cohort of 61 infants (Supplementary Fig. 5b, c), the five class labels (IC1-5) were assigned to the 23 additional cases, which recapitulated the cluster-specific DNA methylation profiles as well as fusion partner distributions (Supplementary Fig. 5d). Moreover, EFS showed a similar trend, with IC2 infants exhibiting the poorest EFS (Supplementary Fig. 5e). Collectively, our dual-omics unsupervised clustering identified five prognosis-relevant ICs in *KMT2A*-r infant ALL.

### Differential enrichment of developmental signatures defines the ICs

Since the addition of methylation information improved the clustering resolution, we next examined the genome-wide DNA methylation status of these five ICs. Globally, DNA methylation levels showed similar distributions between ICs across the genome as well as within gene and promoter loci (Supplementary Fig. 6a, b), suggesting that small sets of IC-specific target loci, rather than genome-wide drastic

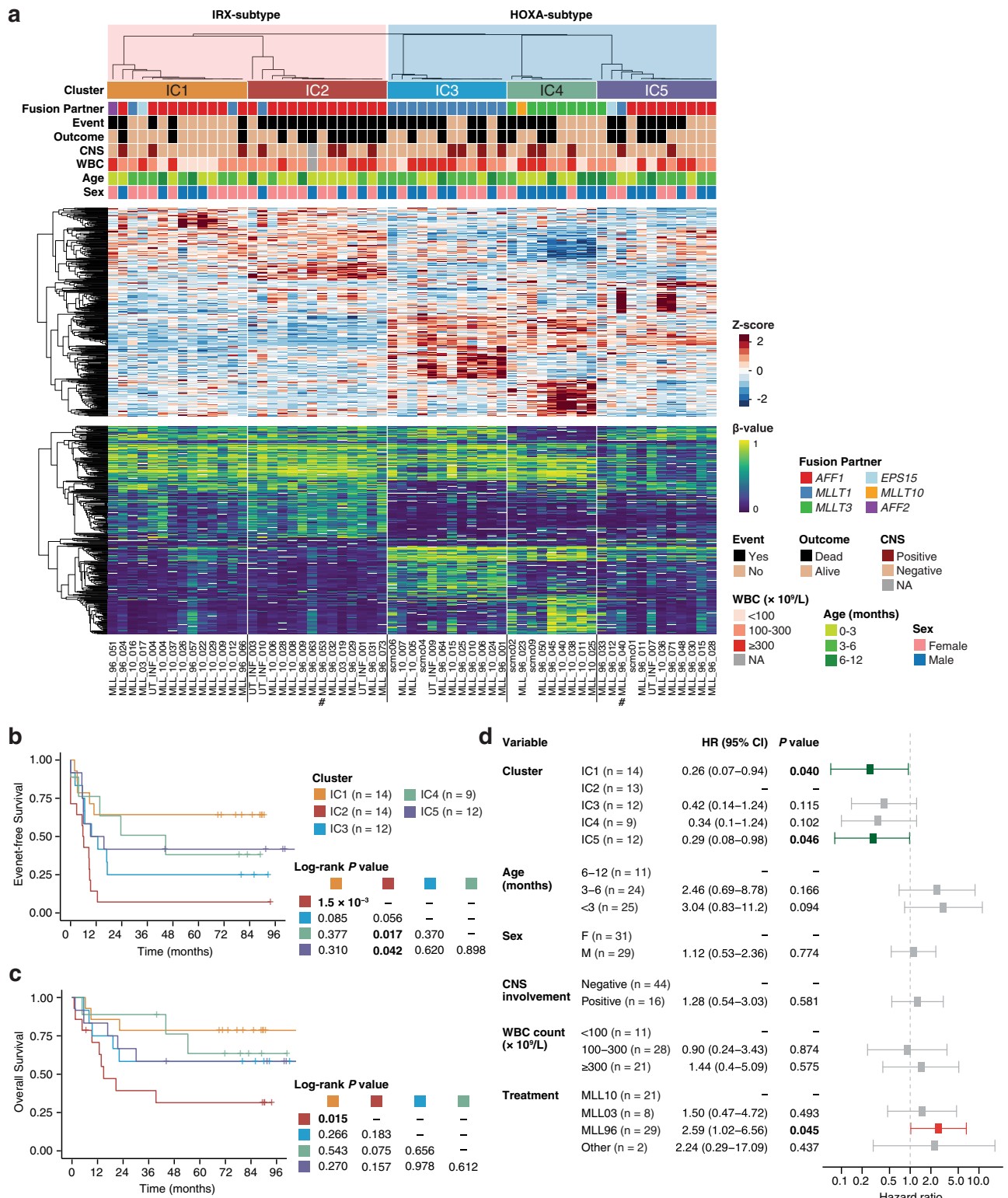

**Fig. 1 | Dual-omics-integrative clustering of *KMT2A*-r infant ALL.**
**a** Comprehensive dual-omics heatmaps of *KMT2A*-r infant ALL. Top: clinicopathological features of the discovery cohort of 61 infants. The dendrogram is from consensus clustering and indicates sample-to-sample distances. Middle: Expression heatmap of cluster marker genes. The top 100 significantly up- or downregulated marker genes for each of the five ICs were included. Bottom: Methylation heatmap of cluster marker probes. The top 100 significantly hyper- or hypomethylated marker probes for each of the five ICs were included. Number signs (#) indicate the two cases with B/M MPAL. CNS central nervous system, WBC white blood cell, NA not available. **b**, **c** Survival analysis comparing the EFS (**b**) and OS (**c**) of different ICs of *KMT2A*-r infant ALL. **d** Multivariate Cox proportional hazards analysis for EFS. ICs and known clinical prognostic factors, as well as treatment protocols, were included in the model. One patient (MLL_96_063) was excluded due to missing clinical information; a total of 60 infants were included, and the subtotal number of cases in each variable group is indicated. Error bars show the 95% confidence interval. Two-sided raw *P* values are shown. HR hazard ratio, CI confidence interval.

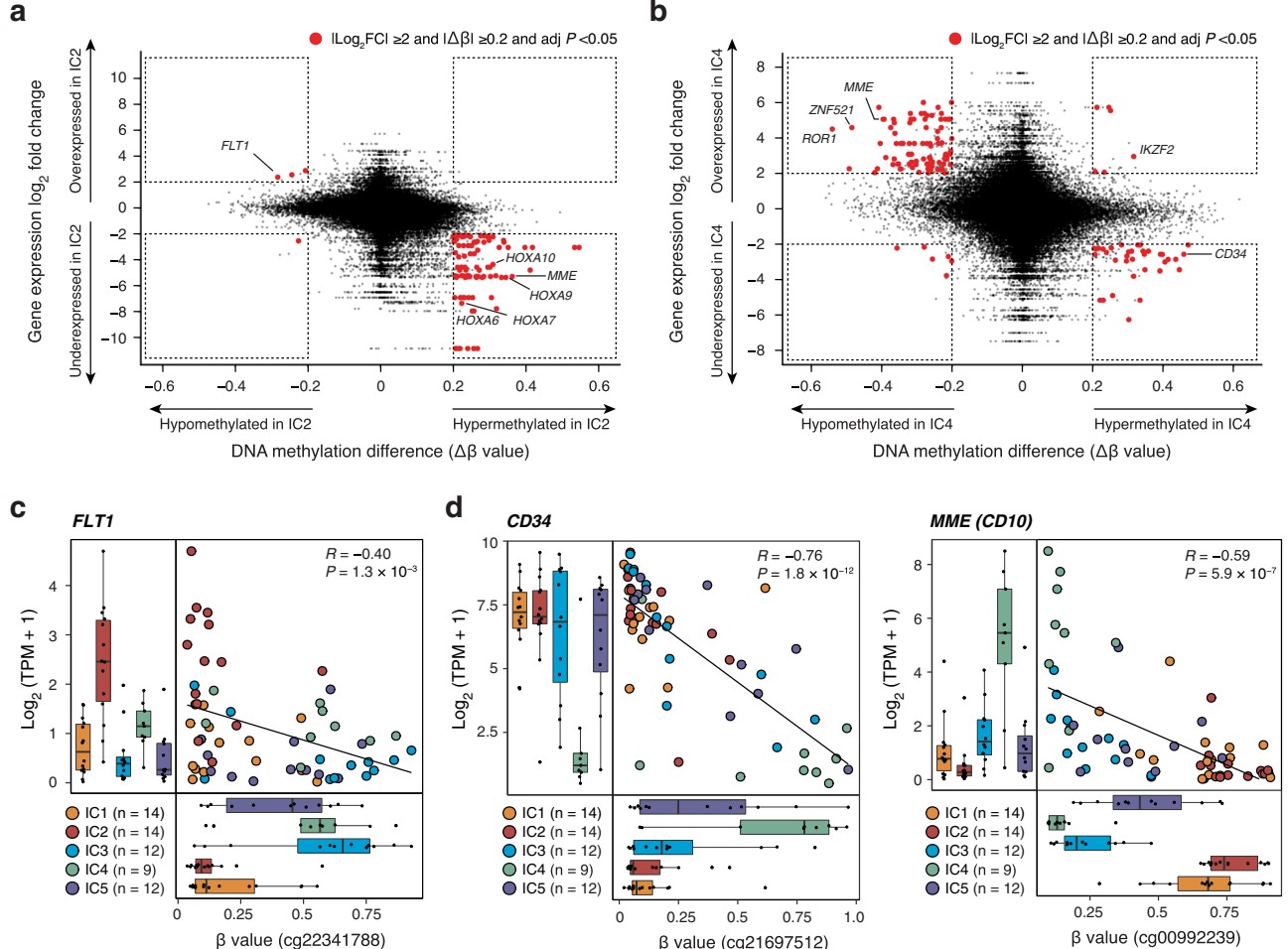

**Fig. 2 | Integrative comparison of transcriptome and DNA methylome in infant ALL. a, b** Integrative scatter plots contrasting expression differences with DNA methylation differences in IC2 (**a**) and IC4 (**b**). Genes and probes with significant differences in gene expression (|$\log_2$ fold change| ≥2 and two-sided adjusted $P < 0.05$) and DNA methylation (|Δβ| ≥0.2 and two-sided adjusted $P < 0.05$) are highlighted in red. **c, d** Correlated gene expression and DNA methylation of dual-omics cluster markers *FLT1* (**c**), *CD34* and *MME* (**d**). Pearson correlation coefficients (*R*) and raw Pearson correlation *P* values are indicated. Box plots show median and first/third quartiles. The whisker extends from the smallest to the largest values within 1.5 × IQR from the box hinges.

differences, distinguish the IC methylomes. To define IC-specific markers, we next jointly analyzed the transcriptome and DNA methylome of our discovery cases. First, expression-based marker genes were identified for each cluster (Supplementary Data 4), which were further narrowed down based on differential methylation status. Consequently, 38, 32, and 73 genes were identified as dual-omics marker genes of IC2, IC3, and IC4, respectively (Fig. 2a, b and Supplementary Fig. 6c and Supplementary Data 5), although no genes were identified as dual-omics markers for IC1 and IC5. Among the dual-omics markers, *FLT1* showed the greatest methylation reduction and concordant upregulation in IC2 (Fig. 2c). *FLT1* encodes vascular endothelial growth factor receptor 1 (VEGFR1), which is not physiologically expressed in hematopoietic progenitors[22] (Supplementary Fig. 6d) but is reported to be responsible for intra-bone marrow localization and survival of ALL cells[23]. Furthermore, key hematopoietic and B-lineage markers, including *CD34, MME* (encoding CD10), and *DNTT*, were found among the dual-omics marker genes (Fig. 2d and Supplementary Fig. 6e), suggesting that different ICs have different developmental status towards the B-cell lineage. In fact, conventional B-cell developmental marker genes were variably expressed and methylated between ICs (Supplementary Figs. 7 and 8).

To further resolve the cluster-specific developmental stages, we utilized the B-lineage cell type markers defined by a single-cell transcriptomics study of human fetal livers (Supplementary Data 6)[22], and evaluated single-sample enrichment of these developmental signatures. Intriguingly, the B-cell developmental signatures were differentially enriched in the five ICs, with distinctive enrichment of the hematopoietic stem cell/multipotent progenitor (HSC/MPP) signature in IC1, the pre–pro-B signature in IC3 and IC5, and the pre-B signature in IC4 (Fig. 3a, b). The enrichment of the pre-B signature in IC4 was consistent with the *CD34*-negative, *MME*-positive expression pattern of this subgroup (Fig. 2d). Comparison with published RNA sequencing[24] and methylation array[25] datasets of normal B-cell progenitors further confirmed the more mature status of IC4 by identifying shared expression and methylation signatures between IC4 and normal progenitors (Supplementary Fig. 9). Of additional interest was the poor-prognosis IC2, which exhibited negative signature enrichment for all B-cell progenitor stages (Fig. 3a, c), even compared with IC1 (Supplementary Fig. 10a), defining this cluster as the most undifferentiated subgroup of infant ALL. Although myeloid marker co-expression and lineage switch are commonly observed in the treatment of *KMT2A*-r infant ALL, expression of myeloid lineage signatures did not differ significantly between ICs (Supplementary Fig. 10b).

Since these results indicated that the IRX subtype (IC1-2) is less differentiated towards the B-cell lineage compared to the HOXA subtype (IC3-5), we hypothesized that IRX-type infant ALL retains the cellular states of earlier hematopoietic development. To address this, we extracted the positive targets (regulons) of *IRX1* and *HOXA9* by

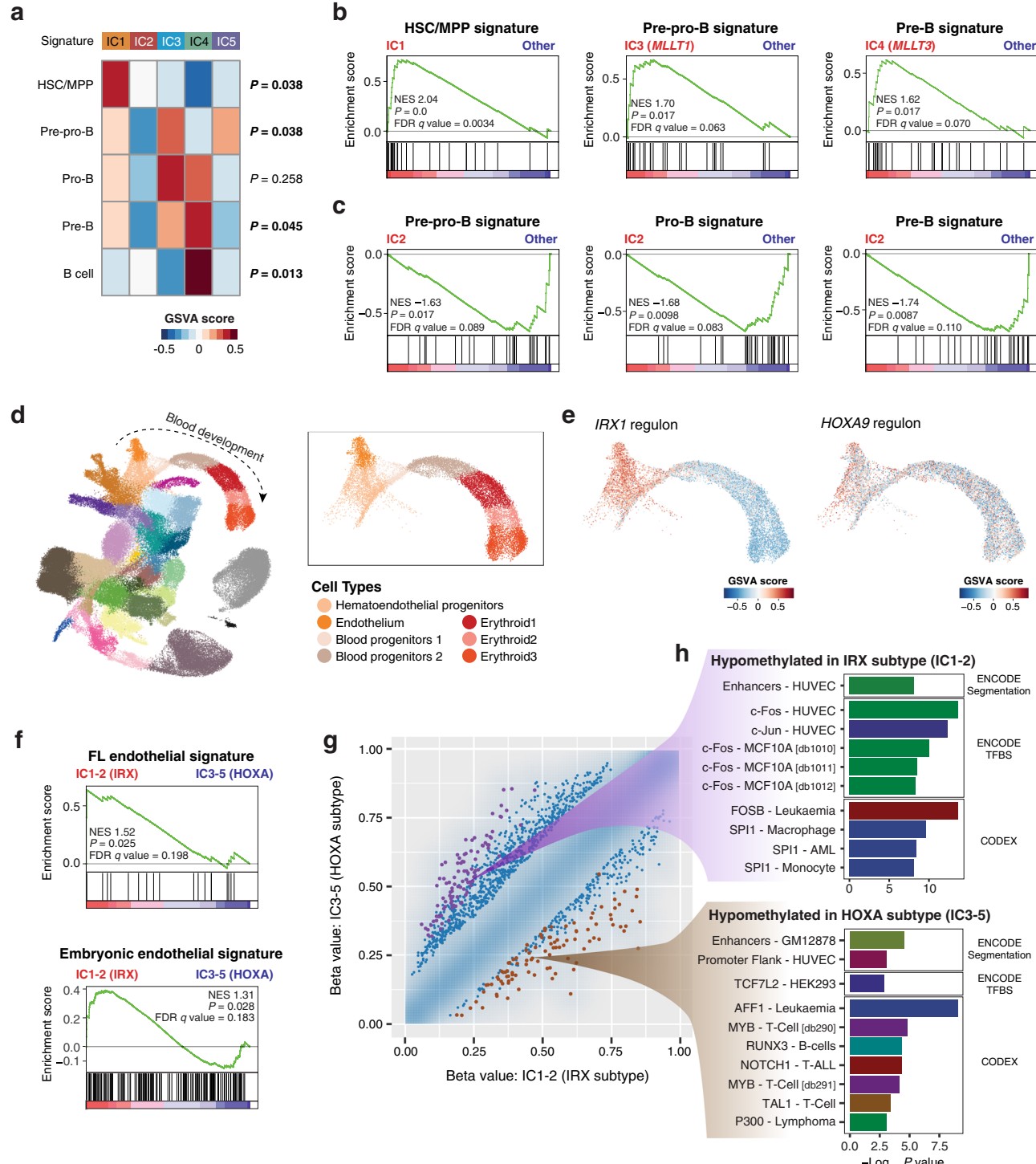

**Fig. 3 | Differential enrichment of B-lineage and hemato-endothelial developmental signatures in *KMT2A*-r infant ALL. a** Enrichment analysis comparing the expression of B-cell developmental signatures in the five ICs. Enrichment scores were calculated at the individual sample level using GSVA, and the median value for each signature in each cluster is plotted. Kruskal–Wallis *P* values are indicated on the right of the heatmap. **b** Cluster-specific enrichment of B-lineage signatures evaluated with GSEA. NES normalized enrichment score, FDR false discovery rate. **c** Significant under-enrichment of B-progenitor signatures in IC2. **d** Uniform manifold approximation and projection (UMAP) visualization of a published single-cell gene expression atlas of murine gastrulation (*n* = 116,312 cells)[26]. The right upper box shows a magnified section of the blood development trajectory

(*n* = 13,881 cells). **e** GSVA enrichment of positive targets of *IRX1* and *HOXA9* within the blood development trajectory. **f** Enrichment analysis of endothelial cell signatures comparing IRX subtype vs. HOXA subtype. **g** Density scatter plot comparing the DNA methylation levels of IRX subtype and HOXA subtype. Differential methylation was assessed on the basis of genomic tiling regions with a 5-kb window. Methylation levels of the tiling regions are shown as density in blue, and individual points are plotted in the 1% sparsest areas of the plot. The 100 most significantly hypomethylated regions in the IRX and HOXA subtypes are shown in purple and brown, respectively. **h** LOLA enrichment analysis on the 100 differentially methylated regions highlighted in (**g**). The top ten most significantly enriched categories from CODEX or ENCODE entries of the LOLA Core databases are shown.

network analysis of patient gene expression data (Supplementary Fig. 1d and Supplementary Table 2), which were then projected onto a transcriptomic atlas of early organogenesis[26] (Fig. 3d). Projection of single-cell activities of the *IRX1* and *HOXA9* regulons revealed that the *IRX1* regulon activity was clearly enriched in hemato-endothelial progenitor and endothelial cell compartments, whereas the *HOXA9* regulon was not enriched in any particular cell type (Fig. 3e). Accordingly, endothelial cell signatures derived from human fetal liver[22] as well as early human embryonic development[27] were enriched in the IRX subtype compared with the HOXA subtype (Fig. 3f), whereas no significant difference was observed between IC1 and IC2 (Supplementary Fig. 10c). DNA methylation status confirmed this relationship, with endothelial enhancers hypomethylated in IRX subtype and lymphoid progenitor enhancers hypomethylated in HOXA subtype (Fig. 3g, h). Altogether, signature activities of B-lineage and early hemato-endothelial development defined the cluster-specific developmental stages of infant ALL.

## Distinct landscapes of genomic alterations in the ICs of *KMT2A*-r infant ALL

To further characterize the molecular basis of infant ALL subgroups, we next explored the mutational landscape of all 61 infants from the discovery cohort using whole-exome sequencing (WES) and/or targeted deep sequencing (deep-seq) of recurrently mutated genes in infant ALL (Supplementary Data 7). First, WES was performed in 19 of the 61 discovery cases with a mean sequencing depth of 140 (Supplementary Fig. 11a, b), by which 63 non-silent somatic single-nucleotide variations (SNVs) or insertions/deletions (indels) were detected, and 59 SNVs/indels in 44 genes (94%) were validated by amplicon or capture-based deep sequencing (Supplementary Data 8). The number of 3.1 non-silent mutations per exome was compatible with the previous reports[8,12], confirming the low mutational burden of infant ALL (Supplementary Fig. 11c). In addition, deep-seq was performed in all 61 discovery cases with an average depth of 998 (Supplementary Fig. 11d), and identified 93 non-silent mutations in 11 genes (Supplementary Data 9). In the 19 cases analyzed with both WES and deep-seq, WES failed to detect 6 of 19 (32%) subclonal mutations with variant allele frequencies (VAFs) <10% identified with deep-seq (Supplementary Fig. 11e), indicating a 1.5-fold higher sensitivity of our deep-seq over WES (19 vs. 13 mutations) for subclonal mutation detection. Furthermore, 29 copy number alterations (CNAs; 1.5 per case) and two *KMT2A*-unrelated structural variations (SVs) were identified using WES and deep-seq (Supplementary Table 3 and Supplementary Data 10).

Combining all mutations and CNAs identified by WES and deep-seq, receptor tyrosine kinase-RAS (RTK-RAS) pathway genes were most frequently altered (Fig. 4a). Among them, *KRAS* (21 of 61 cases; 34%), *FLT3* (20 of 61 cases; 33%), and *NRAS* (19 of 61 cases; 31%) were the most recurrent genes; overall, 44 of 61 cases (72%) harbored at least one mutation in the RTK-RAS pathway. Compared with the previously reported frequencies of RTK-RAS pathway mutations in infant ALL (43–47%)[8,12], our >1.5-fold higher frequency can be explained by the higher sensitivity of our deep-seq for subclonal mutation detection. Indeed, 45 of 93 (48%) mutations detected by deep-seq were subclonal with VAFs <10% (Fig. 4b and Supplementary Data 9).

When comparing the mutational landscapes between clusters, the poor-prognosis IC2 was characterized by the 100% frequency (14 of 14 cases) of RTK-RAS pathway mutations. Strikingly, IC2 was also found to have accumulated a significantly higher number of RTK-RAS pathway mutations than all other ICs (Wilcoxon rank-sum test $P < 0.05$), with a mean of 2.6 mutations (range of 1–6) in each case of IC2 (Fig. 4c). Concordantly, RAS-downstream MAPK and ERK pathways were significantly upregulated in IC2, distinguishing IC2 from the other IRX subtype of IC1 (Supplementary Fig. 10d). Of note, although neither simple positivity of RTK-RAS pathway mutations nor co-occurrence of

*FLT3* and *RAS* mutations was significantly associated with survival rates (Supplementary Fig. 12a–d), upon stratification by the number of RTK-RAS pathway mutations per case, infants with ≥3 mutations in RTK-RAS pathway showed significantly worse EFS compared with infants without RTK-RAS pathway mutations (log-rank $P = 0.028$; Fig. 4d), highlighting the importance of accurate detection of subclonal diversity for risk stratification of infant ALL. Indeed, in one case of IC2 (UT_INF_001) for which a relapse sample was available, the two RTK-RAS mutations (*FLT3* and *PTPN11*) at diagnosis increased the VAFs at relapse (Supplementary Table 4), indicating the contribution of these mutations to relapse in this case, although the significance of RTK-RAS mutations as relapse drivers should be further evaluated with a larger cohort.

Another cluster with a distinct mutational landscape was IC4, which exhibited a significantly higher frequency of CNAs and/or mutations in cell cycle regulators (*CDKN2A/B* and *CCND3*; Fisher's exact $P = 1.1 \times 10^{-3}$) and *PAX5* gene (Fisher's exact $P = 3.9 \times 10^{-4}$; Fig. 4a). In addition, 12 of 23 cases in the extended cohort were subjected to WES and/or deep-seq, whereby 3 of 3 cases in IC2 harbored single dominant (VAF > 40%) or multiple RTK-RAS pathway mutations (Supplementary Figs. 5f and 13 and Supplementary Table 5 and Supplementary Data 11–13). Finally, for additional validation, we exploited a published RNA sequencing dataset of 31 diagnostic samples of *KMT2A*-r infant B-ALL (EGAS00001000246)[8]. By building and applying an expression-based KNN classifier, the 31 infants were assigned with IC labels that recapitulated the cluster-specific expression patterns and characteristic distribution of fusion partners (Supplementary Fig. 14). Again, IC2 was shown to have a significantly higher frequency of RTK-RAS pathway mutations than the other ICs (100% vs. 35%; Fisher's exact $P = 0.012$), although the number of subclonal mutations should be further validated using a deep sequencing method. Overall, our deep sequencing identified conspicuous subclonal diversity in infant ALL, which has been previously reported but remains under-appreciated, and characterized the poor-prognosis IC2 by a significantly higher burden of RTK-RAS mutated subclones.

## KMT2A fusions directly control the IRX/HOXA switch to induce different malignant phenotypes

In spite of the mutually exclusive expression pattern of *IRX* and *HOXA* transcription factors, four cases (6.6%) expressed both *IRX1* and *HOXA9* with transcripts per million (TPM) >10 (Fig. 5a). To better understand the cellular level regulatory mechanisms of the IRX/HOXA axis, we next conducted single-cell RNA sequencing on a double-positive case in IC2 (MLL_96_073). From the diagnostic peripheral blood sample with a leukemia cell content of 95%, viable CD45 dim CD19 + leukemia cells were sorted and sequenced (Fig. 5b and Supplementary Fig. 15a). In accordance with the observation from bulk RNA sequencing, *IRX1* and *HOXA9* were expressed in a mostly mutually exclusive manner at the single-cell level (Fig. 5c and Supplementary Fig. 16a), although a minor fraction (89 of 2004 cells, 4.4%) of double-positive cells should be further investigated for their true existence. Single-cell-level enrichment analysis revealed clusters of cells with differential enrichment of B-cell developmental signatures (Fig. 5d), indicating that leukemic blasts from a single patient contain significant heterogeneity that mimics normal development towards the B-cell lineage. By identifying the cluster marker signatures, each cluster was assigned a corresponding developmental stage (Fig. 5e). Nearest neighbor projection and pseudotime inference based on the fetal liver B-cell progenitors further confirmed the developmental hierarchy within leukemic blasts (Supplementary Fig. 16b–d), where leukemia cells showed sequential expression of developmental markers resembling normal B-cell progenitors (Supplementary Fig. 16e, f), while also exhibiting ectopic early and/or prolonged expression of other developmental regulators (Supplementary Fig. 16g). Considering the HSC/MPP cluster as a leukemic stem cell (LSC) population, *HOXA9* single-positive LSCs showed significant upregulation of proliferative

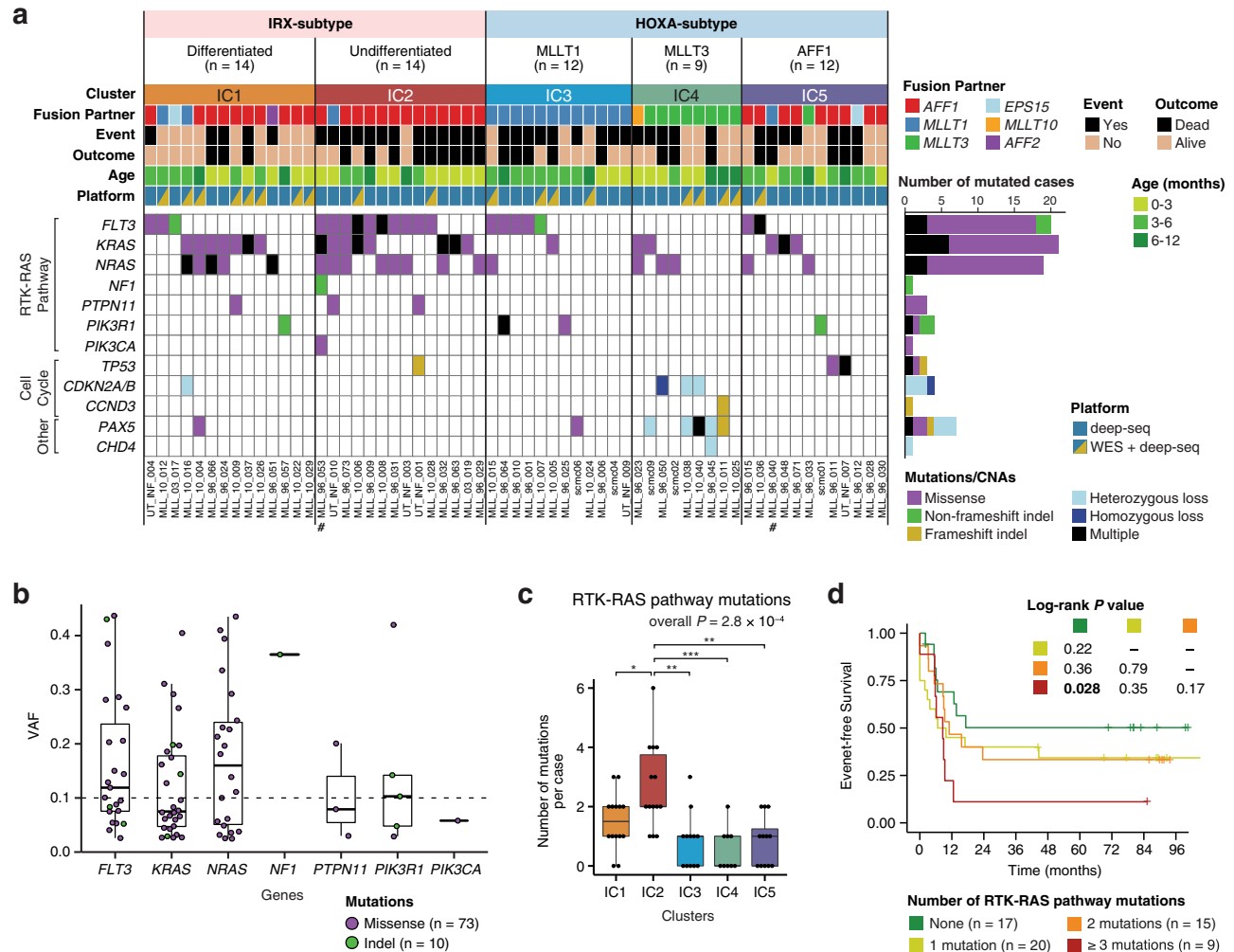

**Fig. 4 | Mutational landscape of *KMT2A*-r infant ALL subgroups. a** Gene mutations and CNAs in the different subgroups of *KMT2A*-r infant leukemia. Number signs (#) indicate the two cases with B/M MPAL. **b** VAFs of RTK-RAS pathway mutations detected with deep-seq. The VAF of 0.1 is indicated by a dashed horizontal line. Box plots show median and first/third quartiles. **c** Number of mutations in the RTK-RAS pathway per case. All 61 cases of the discovery cohort are included: IC1 (*n* = 14), IC2 (*n* = 14), IC3 (*n* = 12), IC4 (*n* = 9), and IC5 (*n* = 12). Box plots show median and first/third quartiles. Overall *P* value is from Kruskal–Wallis test. *P* values for pairwise cluster comparisons are from the two-sided Wilcoxon rank-sum test. \**P* < 0.05; \*\**P* < 0.01; \*\*\**P* < 0.001. Raw *P* values are as follows: *P* = 0.034 (IC2 vs. IC1), 1.2 × 10⁻³ (IC2 vs. IC3), 7.0 × 10⁻⁴ (IC2 vs. IC4), and 1.2 × 10⁻³ (IC2 vs. IC5). **d** Survival analysis based on the number of RTK-RAS pathway mutations.

lymphoid cell signatures as well as cell cycle genes (Fig. 5f), characterizing *IRX1* single-positive LSCs as a more quiescent LSC population. Likewise, a *HOXA9* single-positive case (MLL_96_015) exhibited hierarchical differentiation within leukemic blasts, whose LSC population also exhibited upregulation of activated lymphoid cell markers compared with *IRX1* single-positive LSCs of MLL_96_073 (Supplementary Figs. 15b and 17).

Since these results indicate that the IRX/HOXA molecular switch is strictly controlled at the single-cell level and is associated with different stem cell phenotypes, we next aimed to investigate whether the IRX/HOXA dichotomy is directly governed by KMT2A fusion oncoproteins. To this end, we performed chromatin immunoprecipitation with sequencing (ChIP-seq) for three infant ALL-derived cell lines (IRX-type *KMT2A-AFF1*, HOXA-type *KMT2A-AFF1*, and HOXA-type *KMT2A-MLLT1*)[28,29]. Notably, in the IRX-type cell line (PER-785), ChIP signals of KMT2A, H3K4me3, H3K27ac, and RNA polymerase II were highly enriched in the *IRX1* gene locus but undetectable in the *HOXA* cluster locus, and vice versa in the HOXA-type cell lines (PER-494 and PER-784), consistent with the mRNA expression profiles (Fig. 6). In line with the direct binding of KMT2A to the *IRX1* locus, a reporter assay demonstrated increased activity of the *IRX1* promoter as well as the

*HOXA9* promoter by a chimeric KMT2A-Aff1 fusion (Supplementary Fig. 18a), which recapitulates and models KMT2A-AFF1-driven B-ALL[30], indicating that KMT2A-AFF1 has the potential to directly induce both IRX and HOXA programs. By defining enhancers as H3K27ac-positive and H3K4me3-low or negative loci, enhancers significantly correlated with higher expression of target genes (Supplementary Fig. 18b, c), where HOXA-type enhancers were enriched in more committed B-cell differentiation/activation-related loci compared with IRX-type enhancers (Supplementary Fig. 18d, e). Altogether, these observations suggest that KMT2A fusions, particularly KMT2A-AFF1, have the capacity to choose and activate either the IRX or HOXA program and eventually cause the different developmental phenotypes of B-ALL.

### IRX-type ALL requires lymphoid-unprimed HSCs/MPPs as the origin

Finally, since the molecular profiles of patients and cell lines consistently revealed the distinct B-lineage maturity of IRX-type and HOXA-type infant ALL, we asked whether the cells of origin played a role in the fate decision by KMT2A fusions toward the IRX or HOXA subtype. To investigate the potential to develop IRX-type and HOXA-

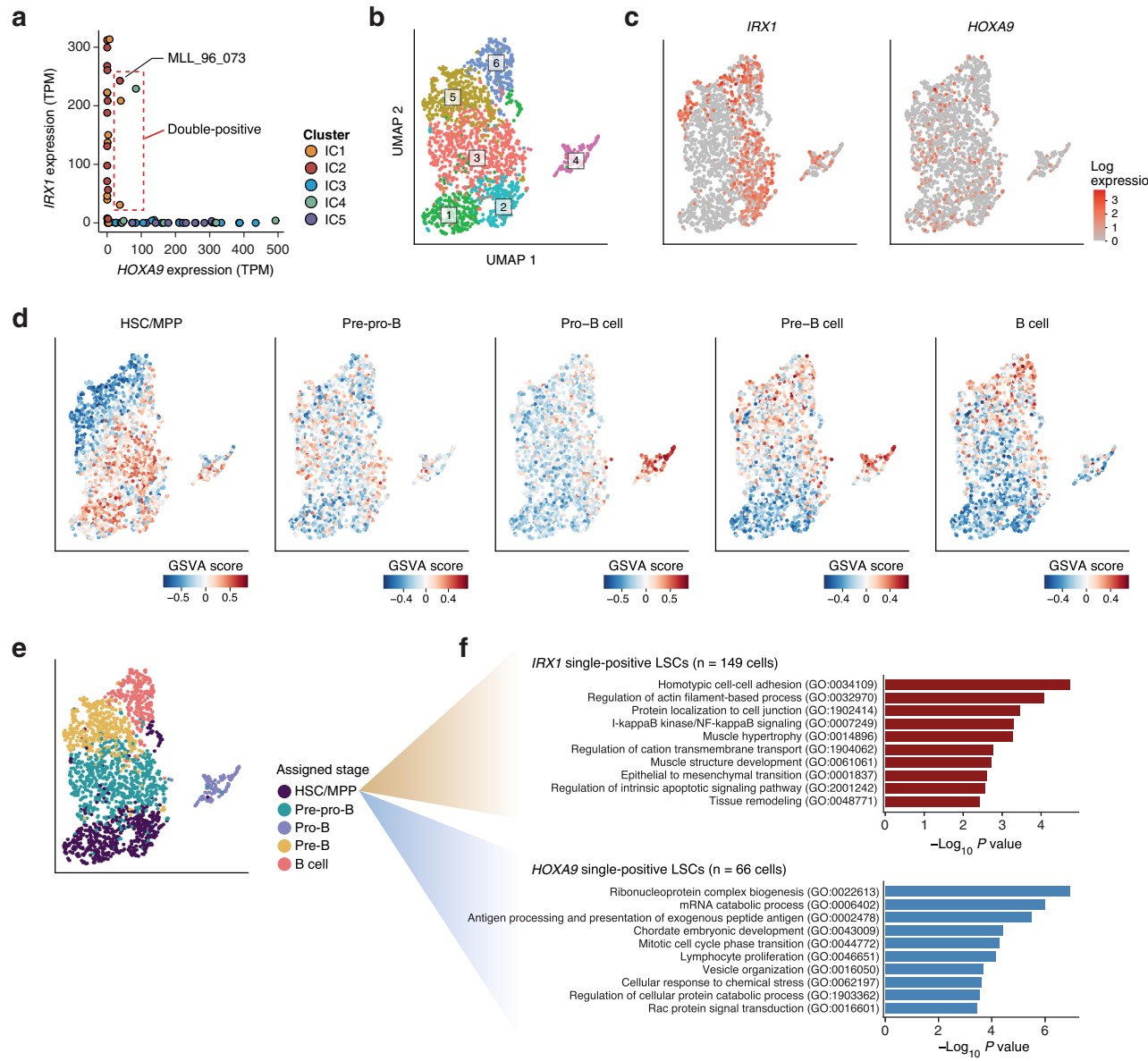

**Fig. 5 | Single-cell transcriptome analysis of an infant with IRX/HOXA double-positive ALL. a** Two-dimensional display of *IRX1* and *HOXA9* gene expression in each patient sample. A case of double-positive ALL (MLL_96_073) is denoted. **b** UMAP visualization of the single-cell transcriptome of MLL_96_073. Clusters were identified with the Louvain method. **c** Log normalized expression values of *IRX1* and *HOXA9* in the leukemia cells of MLL_96_073. **d** Single-cell-level enrichment of B-lineage developmental signatures computed with the GSVA algorithm. **e** Inferred developmental stages of the leukemia cells from MLL_96_073. Marker signatures were computed for the cell clusters identified in (**b**), and were assigned as cluster-defining developmental stages. **f** Enriched Gene Ontology terms in the *IRX1*-positive LSCs (*n* = 149 cells; upper panel) and *HOXA9*-positive LSCs (*n* = 66 cells; lower panel).

type ALL, HSCs, MPPs, lymphoid-primed multipotent progenitors (LMPPs) and common lymphoid progenitors (CLPs) were purified from human umbilical cord blood (CB), transduced with KMT2A-Aff1[30], and cultured in lymphoid-promoting conditions (Fig. 7a and Supplementary Fig. 19a). By week 6, all four hematopoietic stem and progenitor cell (HSPC) compartments were transformed into B-ALL phenotype with CD34⁻ CD19⁺ and variable CD10 expression (Fig. 7b and Supplementary Fig. 19b), including from originally CD10⁺ CLPs. Intriguingly, *IRX1*-positive leukemia blasts emerged only from HSCs or MPPs, whereas *HOXA9*-positive blasts emerged from all four HSPC compartments (Fig. 7c). These results demonstrate that IRX-type leukemogenic potential is lost earlier than HOXA-type potential in the lymphoid developmental trajectory, implicating the cells of origin as a key factor in the IRX/HOXA fate decision by KMT2A fusions.

## Discussion

This study demonstrates the heterogeneous molecular pathobiology of *KMT2A*-r infant ALL using a fully unsupervised, integrative multi-omics approach. As various types of cancers have benefited from omics-integrative approaches for the discovery of clinically relevant novel disease subgroups[15–17], our SNF-based dual-omics clustering robustly identified five subgroups (IC1-5) in *KMT2A*-r infant ALL, which significantly correlate with patient prognosis. Since SNF identifies both consistent and complementary patient-to-patient similarities across multiple omics layers[21], it is most likely that our SNF-based clustering approach successfully integrated the shared and distinct patient similarities captured with transcriptome and methylome information, which were, however, not distinct enough to cluster in a single-omics analysis.

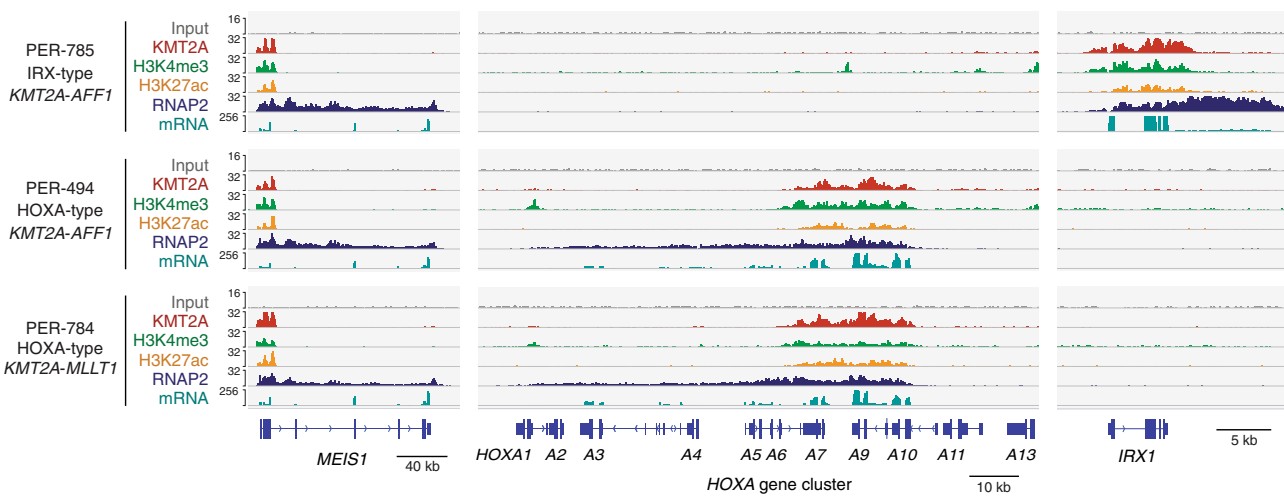

**Fig. 6 | Different chromatin landscapes of IRX and HOXA subtypes of *KMT2A*-r infant ALL cell lines.** ChIP-seq profiles of KMT2A, H3K4me3, H3K27ac, and RNA polymerase II (RNAP2) at the *MEIS1*, *HOXA*, and *IRX1* loci. RNA sequencing profiles are also depicted.

These five ICs are broadly classified into two subtypes: IRX subtype and HOXA subtype. The former was first identified by Trentin and colleagues as infants with *KMT2A-AFF1*-driven ALL lacking the hallmark expression of *HOXA* genes[31]. Over the past decade, several gene expression profiling studies have validated this *HOXA*-low IRX subtype within *KMT2A-AFF1* infant ALL, and its higher relapse rate has also been reported[12,13,20]. Indeed, simple expression-based clustering of our discovery cases confirmed the IRX/HOXA substructure, while also revealing a rare existence of non-*KMT2A-AFF1* cases in this subtype. Of note, however, our omics-integrative clustering further segregated the IRX subtype into two distinct subgroups (IC1 and IC2) with highly discriminative prognoses, which have never been described to date. The most pronounced genomic characteristics of the poor-prognosis IC2 are the 100% frequency of RTK-RAS pathway mutations and the conspicuous intra-sample heterogeneity with a high burden of RTK-RAS mutant subclones. Recently, Ma et al. have shown that the relapse founder clone in pediatric B-ALL often originates from a minor subclone at diagnosis, and that subclonal RAS pathway mutations with low VAFs of ~2% can expand and seed relapse[32]. Therefore, the existence of multiple different RTK-RAS mutant subclones in IC2 may increase the chances that at least one of these mutant subclones will break through treatment and lead to relapse. In fact, our sequencing of a paired relapse sample from a case of IC2 (UT_INF_001) demonstrated that two RTK-RAS mutations at diagnosis had contributed to relapse in this case. However, the disappearance of RTK-RAS mutations at relapse has also been reported[8], indicating that RTK-RAS wild-type subclones can compete and predominate in such cases, necessitating further investigation into the mechanisms of how RTK-RAS mutant subclones contribute to relapse. Furthermore, in accordance with the high burden of RTK-RAS mutations, the RAS-downstream MAPK and ERK pathways were significantly upregulated in IC2 over IC1, suggesting potential applicability of targeted drugs, such as MEK inhibitors, for this deadly subgroup. Importantly, the uncovering of this strikingly high RTK-RAS mutational burden in IC2 as well as the higher frequency of RTK-RAS pathway mutations in our entire cohort than in previous studies (72% vs. 43–47%)[8,12] is attributable to our deep mutation detection method with the mean sequencing depth of ~1000. Because infant ALL is particularly enriched with subclonal RTK-RAS mutations and our results show potential contribution of subclonal diversity to higher relapse rates, accurate recognition of subclonal RTK-RAS mutations based on deep sequencing would be of particular importance for clinical decision-making.

Recent development of single-cell transcriptomics also provides insights into the different cells of origin of the infant ALL subtypes. Using sophisticated gene sets derived from developmental cell atlases[22,26], our analysis uncovered the weakest hemato- and B-lymphopoietic development of the poor-prognosis IC2 as well as the shared enrichment of early hemato-endothelial signatures in the IRX subtype (IC1 and IC2). Together with the underrepresentation of B-cell lineage enhancers in an IRX-subtype cell line, these results suggest more immature cells of origin of the IRX subtype compared with the HOXA subtype. Indeed, the IRX-type leukemia required lymphoid-unprimed HSCs/MPPs for CB transformation by KMT2A-Aff1, in contrast to the HOXA subtype, which could arise from LMPPs or CLPs. Since wild-type and fused KMT2A function in a cellular context-dependent fashion[33-35] and KMT2A-Aff1 enhanced the promoter activity of both *IRX1* and *HOXA9* in the context-independent condition of reporter assays, the originating cellular state may be a critical determinant of the IRX/HOXA decision by the KMT2A-AFF1 oncoprotein. Although CB HSCs/MPPs gave rise to both IRX-type and HOXA-type ALL, given the in utero origin of *KMT2A* rearrangement[6,7] and the different epigenetic signatures between fetal liver and CB HSPCs[36], fetal liver HSCs/MPPs may be promising candidates more susceptible to IRX-type ALL.

Another determinant of molecular profiles in *KMT2A*-r ALL are the fusion partner genes. To date, more than 90 fusion partners have been reported, of which three major partner genes are responsible for nearly 90% of *KMT2A*-r infant ALL: *AFF1* (49%), *MLLT1* (22%) or *MLLT3* (16%)[37]. Our discovery cohort (n = 61) well represented the true cytogenetic distribution, comprising 32 *KMT2A-AFF1* (52%), 16 *KMT2A-MLLT1* (26%), and 9 *KMT2A-MLLT3* (15%) cases. Notably, our unsupervised analysis revealed significant correlations between multi-omics molecular profiles and fusion partners, particularly in the HOXA subtype: MLLT1 (IC3), MLLT3 (IC4), and AFF1 (IC5). In contrast to the IRX subtype, the three HOXA-type ICs showed variable enrichment of B-cell progenitor signatures: pre–pro-B signature in IC3 and IC5, and pre-B signature in IC4. The differing stages of arrested B-lineage differentiation may be attributable to the different instructive capabilities of different KMT2A fusions, as suggested by a reported model system[30], in which genetically concordant human CB HSPCs were transformed into pro-B and pre-B-ALL by *KMT2A-Aff1* and *KMT2A-MLLT3*, respectively. A recent case review reporting the recurrent *KMT2A-MLLT3* rearrangement in extremely rare mature B-ALL patients[38] also suggests an instructive preference of *KMT2A-MLLT3* for B-lineage maturation. In addition, two recent DNA methylation

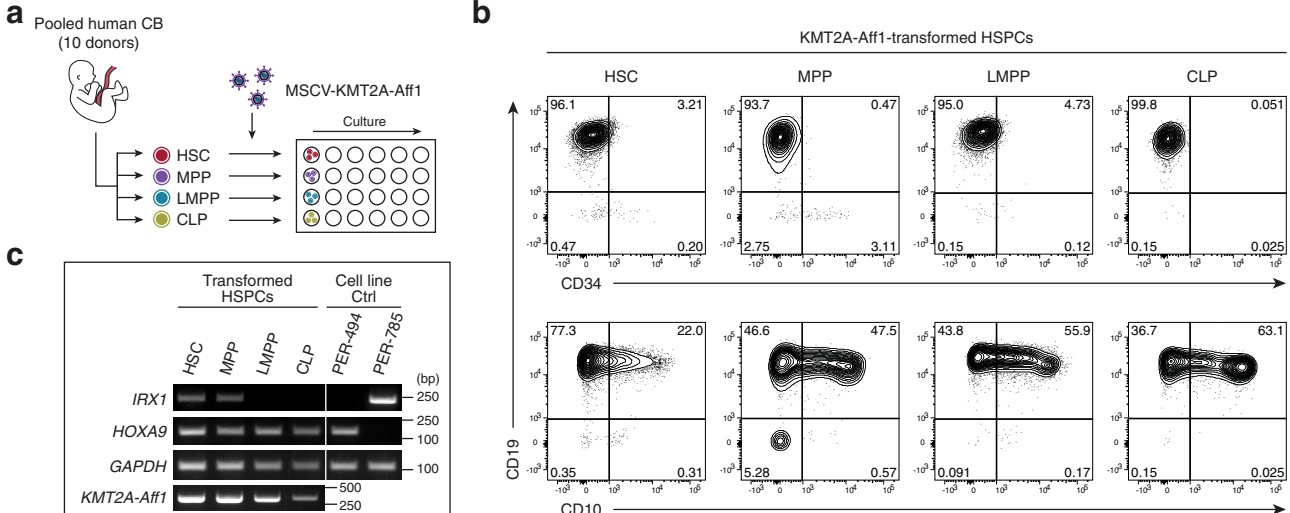

**Fig. 7 | IRX-type leukemia-initiating potential is lost earlier in B-cell development than HOXA-type potential. a** Schematic outline of in vitro CB transformation assay. CB HSPCs were sorted, transduced with KMT2A-Aff1, and cultured on MS-5 stroma cells under lymphoid conditions. **b** Flow cytometry plots of KMT2A- Aff1-transformed HSPCs. **c** RT-PCR of *IRX1* and *HOXA9* in the transformed HSPCs. HOXA-type (PER-494) and IRX-type (PER-785) cell lines are shown as controls. Experiments were independently repeated three times, and representative results are shown. Source data are provided as a Source Data file.

profiling studies demonstrated relatively similar methylation profiles between *KMT2A-MLLT3* and *KMT2A*-germline infants[39,40], which may be due to the common pre-B state of *KMT2A-MLLT3* and *KMT2A*-germline infant ALL. Furthermore, *KMT2A-MLLT3*-driven infant ALL (IC4) exhibited a distinct profile of cooperating genetic alterations with frequent CNAs and mutations in *PAX5* and cell cycle regulators (*CDKN2A/B* and *CCND3*), indicating a unique pathogenesis of this subgroup among *KMT2A*-r infant ALL.

In summary, our multi-omics analysis and functional validation comprehensively illustrate the molecular heterogeneity of *KMT2A*-r infant ALL, represented by distinct mutational landscapes, corresponding stages of arrested normal differentiation, and potential cells of origin. Of particular note, the highest RTK-RAS mutational burden hallmarks the most immature and refractory cases, arguing for the importance of prospective genomics-based stratification and potential therapeutic strategies targeting this pathway, although further preclinical studies are warranted.

## Methods

### Patients and samples

This study comprised 84 infants (aged < 12 months) with B-ALL (*n* = 82) or B/M MPAL (*n* = 2). Written informed consent was obtained from the parents and/or legal guardians in accordance with the Declaration of Helsinki. The research protocol was approved by the Human Genome, Gene Analysis Research Ethics Committee of the University of Tokyo (G0948-(19)), the Ethics Committee of Kyoto University Graduate School and Faculty of Medicine (G-1030-8), the Review Board of Tokyo Medical and Dental University (G2000-193 and G2000-103) and the Review Board of Japan Pediatric Leukemia Study Group (JPLSG) (041). Compensation was not provided to patients. Diagnostic bone marrow or peripheral blood samples were procured from the Japan Children's Cancer Group (JCCG) and its predecessor, the Japanese Pediatric Leukemia/Lymphoma Study Group (JPLSG), as well as from Saitama Children's Medical Center and the University of Tokyo Hospital. The patients were treated according to MLL-10 (*n* = 33)[4], MLL03 (*n* = 12)[41], MLL96/98 (*n* = 37)[42], or other protocols (*n* = 2). In all 84 patients, *KMT2A* rearrangements had been detected by Southern blotting, FISH and/or RT-PCR for treatment stratification. Detailed clinical information for each case and a summary of the experimental design are provided in Supplementary Data 1.

### Bulk RNA sequencing

For the diagnostic leukemia samples from 61 infants for which high-quality RNA (RNA integrity number > 5.0) was available, libraries for RNA sequencing were prepared using the NEBNext Ultra RNA Library Prep kit for Illumina (New England BioLabs) or the TruSeq RNA Library Preparation Kit v2 (Illumina) and were sequenced using an Illumina Hiseq 2000 or 2500 platform with 100–150 bp paired-end mode. Sequencing reads were mapped to GRCh37, and fusion transcripts were called using Genomon (https://genomon-project.github.io/GenomonPagesR/) and filtered by excluding (i) fusions with junctions not located at known exon-intron boundaries; (ii) out-of-frame fusions; (iii) *KMT2A*-unrelated fusions with ≤10 spanning reads. Known *KMT2A* fusions missed by Genomon were restored if identified by Pizzly (https://github.com/pmelsted/pizzly). All candidate fusion transcripts were subjected to experimental validation by RT-PCR and/or Sanger sequencing, except for two cases (MLL_10_021 and MLL_10_036), for which sufficient amounts of RNA samples were not available for validation. In-frame reciprocal *KMT2A* fusions and/or splicing variants of *KMT2A* fusions predicted by genomic structural variations but not bioinformatically identified by RNA sequencing were examined by RT-PCR and were added if proven. Overall, 85 fusions were identified by Genomon, and 21 and 5 fusions were restored by Pizzly and SV-based prediction, respectively (Supplementary Data 3).

### Gene expression analysis

Read counts for each gene were quantified using Genomon, and TPM were calculated. Genes on sex chromosomes and low-expression genes with TPM < 1.0 in 100% of samples or mean TPM < 0.5 were then excluded. Regularized logarithm (rlog) transformation and differential expression analysis were performed using DESeq2 (ref. 43). Two-group comparisons were performed using the Wald test, and *P* values were adjusted with the Benjamini-Hochberg procedure. Multi-class comparisons were performed using the Kruskal–Wallis test. Transcription factor regulatory networks were reconstructed from the rlog-transformed gene-sample expression matrix and putative master regulators and their regulons were computed using the R package RTN[44]. For expression data of normal B-cell progenitors, raw sequencing files of fetal bone marrow RNA sequencing (GSE122982)[24] were downloaded and analyzed using Genomon.

## Methylation array

DNA methylation profiles were analyzed using Infinium Human-Methylation450 (450 K) or MethylationEPIC BeadChip (Illumina), according to the manufacturer's protocol. Raw signal intensity data were processed using the R package RnBeads[45]. Patient methylome data from 450 K and EPIC arrays were first combined using the common probes and preprocessed using NOOB background subtraction and beta-mixture quantile normalization methods. $M$-value matrices from 450 K and EPIC arrays were transformed using the remove-BatchEffect function from the R package limma for unsupervised analyses. For supervised differential methylation analysis, the array platforms (450 K or EPIC) were specified as covariates and were adjusted when calculating the $P$ values. Enrichment of regulatory elements in differentially methylated regions was identified using the R package LOLA[46] with the default parameters as implemented in RnBeads. For methylation data of normal B-cell progenitors, raw signal intensity data from a methylation array study of fetal bone marrow (GSE45459)[25] were downloaded and analyzed using RnBeads.

## Dual-omics-integrative clustering

Dual-omics integration and clustering were performed using SNF[21] and consensus clustering[47] algorithms. The most variably expressed genes ($n = 200$) and the most variably methylated probes ($n = 1000$) were extracted based on the median absolute deviation (MAD) of the rlog values (expression) and $M$-values (methylation), respectively. The rlog and $M$-value matrices were then z-score standardized, from which sample-to-sample Euclidean distances were calculated for each omics layer. These intra-omics distance matrices were converted to affinity matrices and fused to the final similarity network using the R package SNFtool with the following options: alpha = 0.5, $K = 10$, $t = 20$.

Consensus clustering was then performed on the fused dual-omics similarity matrix using the R packages ConsensusClusterPlus and CancerSubtypes[48] with the following parameters: reps = 1000, pItem = 0.8, clusterAlg = "spectralAlg", finalLinkage = "average", distance = "pearson". On the basis of cophenetic correlation coefficient, the proportion of ambiguous clustering scores and silhouette scores, five integrative clusters were identified.

## Gene set enrichment analysis

Single-sample-level gene set enrichment was computed using the R package GSVA with the default parameters[49]. Differential gene set enrichment between patient clusters was evaluated using Kruskal−Wallis test, and cluster median values were used for heatmap visualization. Two-group comparisons were performed using the DESeq2-derived $-\log_{10}$ adjusted $P$ values with the signs of $\log_2$ fold changes as input for preranked GSEA.

A publicly available single-cell mouse gastrulation atlas[26] was used for projection of *HOXA9* and *IRX1* regulon activities in the early development of the hematopoietic system. For this purpose, the RTN-derived *HOXA9* and *IRX1* regulons were converted to mouse gene orthologs using the Ensembl Biomart database v.75 (ref. [50]). Cells within the hemato-endothelial trajectory ($n = 13,881$ cells) were obtained from the single-cell atlas and *HOXA9*/*IRX1* regulon enrichment was calculated using GSVA.

## KNN model building for class prediction of the extended cohort

A KNN classifier was modeled using the KNNXValidation and KNN modules from GenePattern[51] (Broad Institute). First, the hyper-parameters were optimized by leave-one-out cross-validation on the methylation $M$-values or expression $\log_2(\text{TPM} + 1)$ values of the discovery cohort ($n = 61$) with the following options: feature selection statistic = $T$ test, weighting type = distance, distance measure = Euclidean Distance. The hyper-parameter sets were screened for all combinations of the following parameters: num neighbors ($K$) = 3, 5, or 10; num features = 3, 4, 5, 10, 20, 30, 50, 100, 250, 500, 1000, 2500,

5000, or 10000. The pair of ($K$, num features) = (10, 1000) was selected for both methylation and expression classifiers for their best predictive accuracies of 80.3% and 90.2%, respectively. With this set of parameters, the final methylation- and expression-based models were trained with all 61 samples of the discovery cohort, and the class labels were predicted for the extended cohort of 23 infants and a published RNA sequencing cohort[8], respectively.

## Whole-exome sequencing

WES was performed in the cases for which paired tumor and normal DNA samples were available. For the paired normal samples, CD3 + T cells were sorted from the diagnostic samples using MACS beads (Miltenyi). Isolated CD3 + T cells were expanded using T Cell Activation/Expansion Kit (Miltenyi), and DNA was extracted using QIAamp DNA minikit (QIAGEN). DNA libraries were prepared using SureSelect-XT Human All Exon V5 + lncRNA (Agilent Technologies) according to the manufacturer's protocol. Enriched exome libraries were sequenced on an Illumina Hiseq 2000 or 2500 using 100 bp paired-end mode. Sequencing reads were aligned to GRCh37, and SNVs and indels were called by Genomon using the Empirical Bayesian Mutation Calling (EBCall) algorithm[52] with the following parameters: (i) Mapping Quality Score ≥20; (ii) Base Quality Score ≥15; (iii) depths in both tumor and normal ≥8; (iv) number of variant reads in the tumor ≥4; (v) VAFs in tumor samples ≥0.04 for SNVs and ≥0.10 for indels; (vi) VAFs in normal samples <0.02; (vii) EBCall $P$ value ≤$10^{-5}$; and (viii) Fisher's exact $P$ value ≤$10^{-2}$. For mutation detection in known hotspot regions[53], less stringent parameters were used: (i) mapping Quality Score ≥20; (ii) Base Quality Score ≥15; (iii) depths in both tumor and normal ≥8; (iv) number of variant reads in the tumor ≥4; (v) VAFs in tumors ≥0.02; (vi) VAFs in normal samples <0.02; and (vii) Fisher's exact $P$ value ≤$10^{-1.5}$. Following non-hotspot candidates were further excluded: (i) synonymous and ambiguous (unknown) variants; (ii) variants out of coding regions or splice sites; and (iii) variants which were read only from one direction.

## Experimental validation of candidate somatic mutations

All candidate SNVs and indels identified by WES were subjected to validation by PCR-based amplicon deep sequencing and/or capture-based targeted deep sequencing. Among the variants identified with deep-seq, all subclonal SNVs/indels with VAFs ≤0.10 were subjected to PCR-based amplicon deep sequencing for validation. For PCR amplification, a NotI restriction site was attached to each primer as a linker sequence. Amplified products were digested with NotI, ligated, fragmented, and then used for deep sequencing library preparation[54]. In total, 90 of 96 (94%) WES-based candidate mutations and 50 of 50 (100%) deep-seq-based subclonal mutations were validated across the discovery and extended cohorts. The diagnostic and relapse samples of UT_INF_001 were also examined by amplicon deep sequencing.

## Targeted deep sequencing

A custom RNA bait library was designed to capture (i) all coding exons of 30 genes reported to be recurrently mutated in infant ALL[8,12]; (ii) two breakpoint cluster regions (chr11:118350954-118361910; chr11:118367083-118370017) of *KMT2A* gene[55]; (iii) 1250 single-nucleotide polymorphisms (SNPs) for the measurement of genome-wide allele-specific copy numbers; and (iv) additional 95 SNPs for the measurement of copy numbers at 15 gene loci where recurrent CNAs were reported in infant ALL[8] (Supplementary Data 7).

Sequencing libraries were constructed from 200 ng of leukemia genomic DNA samples using the SureSelect-XT Low Input Reagent Kit (Agilent Technologies) according to the manufacturer's protocol. After hybridization capture, the enriched DNA fragments were sequenced on an Illumina HiSeq X Ten platform using 150-bp paired-end mode. Sequence alignment and mutation calling were performed using Genomon with the following parameters: (i) Mapping Quality

Score ≥20; (ii) Base Quality Score ≥15; (iii) sequencing depths ≥8; (iv) number of variant read pairs in the tumor ≥4; (v) VAFs in tumor samples ≥0.02; and (vi) EBCall $P$ value ≤$10^{-4}$. The following were further excluded: (i) synonymous or splice site mutations; (ii) variants which were read only from one direction; (iii) SNVs listed in public SNP databases (NCBI dbSNP build 131 or Human Genetic Variation Database) or our in-house SNP database; (iv) mutations within UCSC simple repeat regions; (v) variants detected in paired normal samples in the WES cohort; and (vi) variants with 0.45 ≤VAF ≤ 0.55 in copy neutral regions, unless confirmed as somatic by paired WES. All final mutation calls were visually inspected with the Integrative Genome Viewer (IGV; Broad Institute) and annotation errors were corrected.

**Detection of copy number alterations and structural variations**
CNAs and SVs were evaluated from WES and/or TCS data. Sequencing-based copy number analysis was conducted using our in-house pipeline CNACS (https://github.com/papaemmelab/toil_cnacs)[56]. CNAs involving sex chromosomes were not evaluated from TCS data because of the sparse distribution of evaluable probes. In samples for which both WES and TCS were conducted, the edges of overlapping segments were determined based on the agreement of both calls and the density and distribution of probes around the edges. Non-overlapping segment calls were merged when WES-based segments and TCS-based segments were adjacent over regions where no probes were placed. For these merged segments, estimated copy numbers were indicated as ranges between WES-based copy numbers and TCS-based copy numbers. SV detection was carried out using Genomon. Putative SVs identified by WES were validated by PCR and Sanger sequencing.

**ChIP sequencing**
Chromatin fractions of infant ALL cell lines (PER-494, PER-784, PER-785) were prepared from $1 \times 10^8$ cells per cell line using the fanChIP method[57]. Briefly, cells were first lysed with lysis buffer to remove chromatin-unbound materials. The chromatin-containing fraction was subsequently treated with micrococcal nuclease (Sigma-Aldrich) for chromatin fragmentation. The chromatin fraction was then subjected to IP with specific antibodies: KMT2A (N-terminal, 14689, Cell Signaling, 1:400), H3K4me3 (39159, Active Motif, 1:400), H3K27ac (308-34843, MABI, 1:400) and RNAP2 (05-623, Millipore, 1:400). Sequencing libraries of the precipitated DNA were prepared using a TruSeq ChIP Sample Prep Kit (Illumina) and sequenced on an Illumina HiSeq 2500 Platform. For each cell line, two independent chromatin preparation and subsequent sequencing were performed (technical replicates; $n = 2$). Sequencing data were aligned using Bowtie2 and replicate consensus peaks were called using HOMER[58] with the $q$-value cutoff of 0.05. Putative cell type-specific enhancers were defined as H3K27ac-positive and H3K4me3-low/negative regions and called using the HOMER getDifferentialPeaksReplicates.pl program. Peaks were visualized using IGV after the aligned sequencing coverages were averaged between replicates. Gene set enrichment was evaluated from the peak calls using the ChIP-Enrich web interface[59] with GO Biological Process annotation and the Chip-Enrich method.

**Single-cell RNA sequencing**
Libraries for single-cell transcriptome sequencing of diagnostic infant leukemia samples (MLL_96_015 and MLL_96_073) were prepared using the BD Rhapsody WTA Amplification Kit (BD Biosciences) and BD Single-Cell Multiplexing Kit (BD Biosciences), according to the BD Rhapsody System mRNA Whole Transcriptome Analysis and Sample Tag Library Preparation Protocol (BD Biosciences). Briefly, CD45$^{dim}$ CD19$^+$ viable leukemic blasts were flow cytometry sorted and labeled with sample tags, and then approximately 5000 cells per were pooled and subjected to single-cell capture and cDNA synthesis with the BD Rhapsody Express Single-Cell Analysis System (BD Biosciences).

Subsequently, cDNA libraries were prepared using the BD Rhapsody WTA Amplification Kit (BD Biosciences) and sequenced on an Illumina HiSeq X Ten platform.

Raw sequencing data was preprocessed using the standard Rhapsody analysis pipeline (BD Biosciences) on the Seven Bridges Platform (https://www.sevenbridges.com). In short, read pairs with low sequencing quality were first removed and the quality-filtered R1 reads were annotated to identify the cell label section sequences and unique molecular identifiers (UMIs). Subsequently, the R2 reads were aligned to the reference sequence using Bowtie2, and valid R1/R2 read pairs were collapsed into raw molecular counts. The raw molecular counts were further corrected by recursive substation error correction (RSEC), an algorithm developed by the manufacturer to correct PCR and sequencing errors. Putative cells with many more reads than noise cell labels were then identified using RSEC-adjusted molecular counts by determining the cutoff point at the minimum second derivative along the cumulative reads curve. All preprocessing parameters used were the default values in the manufacturer's analysis pipeline.

After preprocessing, RSEC-adjusted molecular counts were loaded using the R package Seurat[60]. To retain only high-quality cells, we then removed cells with <1000 read counts, <500 or >3000 genes detected or >15% of UMI counts in mitochondrial genes. Genes detected in <10 cells and mitochondrial genes were also removed to keep only reliable genes. After putative multiplets were further removed using the R package DoubletFinder, 2849 and 2004 cells with a total of 12,306 and 11,796 genes were evaluable for MLL_96_015 and MLL_96_073, respectively. The feature-barcode matrices were then log normalized, and principal component analysis was performed on the 2000 most variably expressed genes. Clustering was performed using the Louvain graph-based algorithm on the first 10 principal components with a resolution of 0.5. Uniform Manifold Approximation and Projection (UMAP) was used for data visualization with the following parameters: umap.method = "umap-learn", metric = "correlation", n.neighbors = 5, min.dist = 0.3.

Developmental signature enrichment was calculated using the GSVA algorithm, and the estimated B-lineage developmental stages were identified for each cluster by using the FindMarkers function on the GSVA enrichment score matrix. Basically, the most significantly upregulated marker signature was assigned as the developmental stage of each cluster. The HSC/MPP stage designation was assigned when the HSC/MPP signature was the only significant marker for the cluster. For pseudotime analysis, B-lineage progenitors (i.e., HSC/MPP, Pre pro-B, Pro-B, Pre-B, and B-cell clusters) were extracted from the fetal liver atlas dataset (https://developmentcellatlas.ncl.ac.uk/datasets/hca_liver/data_share/)[22] based on their original cell type labels. The expression counts were log normalized and 2000 highly variable genes were identified, from which 46 known cell cycle phase markers[61] were further removed. Subsequently, a neighborhood graph was computed with the 1954 highly variable genes (scanpy pp.neighbors with n_neighbors = 5, n_pcs = 10) and pseudotime was calculated with scanpy tl.dpt. Based on the same 1954 genes, differentiation pseudotime of leukemia cells were estimated using Seurat FindTransferAnchors and TransferData (reduction = "cca"). Nearest neighbor-based projection[62] of leukemia cells was performed using the first 10 principal components. Comparison between *IRX1*-positive and *HOXA9*-positive stem cells were performed using the FindMarkers function with the option test.use = "DESeq2", and the top 100 positive markers for each population were used as input to Metascape (https://metascape.org/) to extract significantly enriched Gene Ontology terms.

**Cells and cell culture**
Infant ALL-derived PER-494, PER-784 and PER-785 cells were established as previously described[28,29] and provided by Dr. R.S. Kotecha and

Dr. M.N. Cruickshank. MS-5 cells were established as previously described[63] and provided by Dr. Kazuhiro J Mori. HEK293T cells were purchased from ATCC (Cat #CRL-11268). HEK293T cells were cultured in D-MEM containing 10% fetal bovine serum (FBS). All cell lines used in this study were authenticated by short-tandem repeat analyses and were tested negative for mycoplasma contamination. Human CB cells were obtained from the Japanese Red Cross Kanto-Koshinetsu Cord Blood Bank (Tokyo, Japan) following an institutional review board-approved protocol. Informed consent was obtained in accordance with the Declaration of Helsinki. CB cells from ten different individuals were pooled to reduce experimental batch effects. $CD34^+$ cells were separated using CD34 MicroBead Kit (Miltenyi Biotec). For stem culture, $CD34^+$ cells were cultured in StemSpan (StemCell Technologies) containing 50 ng/ml human stem cell factor (SCF), FLT3 ligand (FLT3-L) and megakaryocyte growth and development Factor (TPO; R&D Systems), and 750 nM StemReginin 1 (ref. 64). For lymphoid culture, $CD34^+$ CB cells and *KMT2A-Aff1* transduced CB cells were cultured on MS-5 stroma cells in aMEM with 20% FBS and supplemented with 10 ng/ml human SCF, FLT3-L and IL-7 (R&D Systems). All experiments using human CB cells were approved by the Ethics Committee at the Institute of Medical Science, the University of Tokyo (approval number: 27-34-1225).

## Vector construction and retrovirus production
Aff1 cDNA was ligated to 5′ KMT2A (Residue 1362) and cloned into the pMSCV neo vector (Clontech). Retroviruses for human cells were produced by transient transfection of 293T cells with viral plasmids along with envelope RD114 and the gag-pol M57 plasmids using the calcium-phosphate method. Retrovirus transduction to the cells was performed by spin infection at 3500 rpm for 4 h.

## Flow cytometry
Cells were stained by fluoro-conjugated antibodies for 30 min at 4 °C. After staining, cells were washed with cold PBS for several times, and were resuspended with PBS containing 2% FBS. Cells were analyzed on a FACSCanto II, a FACSAria III or a FACSAria Fusion using the FACSDiva software (BD Biosciences). Analyses were performed using FlowJo (BD). Dead cells were eliminated by DAPI (BioLegend, catalog #422801) or 7-AAD (Miltenyi Biotec, 130-111-568). The following fluoro-conjugated antibodies were used in this study at indicated dilutions: anti-human Lineage Cocktail BV510 (BioLegend, 348807, clone OKT3; M5E2; 3G8; HIB19; 2H7; HCD56, 1:100), anti-human CD34-APCCy7 (BioLegend, 343513, clone 581, 1:500), anti-human CD38-PECy7 (BioLegend, 303515, clone HIT2, 1:500), anti-human CD90-biotin (BioLegend, 328106, clone 5E10, 1:500), anti-human CD45RA-FITC (BioLegend, 304105, clone HI100, 1:100), anti-human CD10-PE (BioLegend, 312204, clone HI10a, 1:100 for sorting and 1:500 for analysis), Streptavidin-BV605 (BioLegend, 405229, 1:500), anti-human CD34-PECy7 (BioLegend, 343616, clone 561, 1:500), anti-human CD19-APC (BioLegend, 302212, clone HIB19, 1:500), anti-human CD19-APC (Miltenyi Biotec, 130-091-248, 1:33), anti-human CD45 VioBlue (Miltenyi Biotec, 130-113-122, clone 5B1, 1:33).

## RNA extraction and reverse transcription PCR
Total RNA was extracted using the RNeasy Mini kit (Qiagen). RNA was reverse transcribed using Omniscript RT kit (Qiagen), and complementary DNA (cDNA) was then subjected to RT-PCR. Sequences of the primers used for RT-PCR in this study, from 5′ to 3′ are as follows: CGGTACGCGACAACTCTCTG (*IRX1* forward), GGACCGAAGAAGCCC CTTTC (*IRX1* reverse), TCCCTGACTGACTATGCTTGTG (*HOXA9* forward), CAGTTGGCTGCTGGGTTATT (*HOXA9* reverse), TTGAGGT CAATGAAGGGGTC (*GAPDH* forward), GAAGGTGAAGGTCGGAGTCA (*GAPDH* reverse), GCAAACAGAAAAAAGTGGCTCCCCG (*KMT2A* forward), GCTGGAGCTGCTCTCACTCTCA (mouse *Aff1* reverse).

## Luciferase assay
293T cells were seeded in 24-well culture plates at a density of $8 \times 10^4$ cells per well. At 16 h after seeding, the cells were transfected with Firefly Luciferase [FLuc] expressing pGL4.1 *HOXA9* promoter (chr7:27205107-27207332) or pGL4.1 *IRX1* promoter (chr5:3595175-3597779), pGL4.71 vector (expressing Renilla Luciferase [RLuc]) and the KMT2A-Aff1 fusion construct using PEI. The cells were harvested 48 h after transfection, and luciferase activity was measured using the Dual-Luciferase Reporter Assay System (Promega) and a FLUOstar OPTIMA luminometer (BMG LABTECH). Promoter activity was inferred as the ratio of Fluc to Rluc.

## Survival analysis
Overall survival was calculated from the date of diagnosis to the date of death from any cause. Event-free survival was defined as the time period from the date of diagnosis until the date of the first event (failure to achieve remission (calculated as an event at time = 0 d), relapse, secondary malignancy, or death from any cause) or the date of last follow-up. Survival was estimated using the Kaplan–Meier method, and the difference was tested using the log-rank test. For multivariate analysis, a Cox proportional hazards regression model was used to identify the risk factors associated with the EFS rate. The model included cluster, age, sex, WBC count, CNS involvement, and treatment protocol as covariates. *P* values of less than 0.05 were considered statistically significant. Statistical analysis was performed using the R package survival.

## Reporting summary
Further information on research design is available in the Nature Research Reporting Summary linked to this article.

## Data availability
All RNA sequencing, methylation array, whole-exome sequencing, targeted deep sequencing, ChIP sequencing, and single-cell RNA sequencing data generated in this study have been deposited in the Japanese Genotype-phenotype Archive (JGA) under accession code JGAS000385. Access can be requested through the National Bioscience Database Center (NBDC) application system as detailed in the instructions [https://humandbs.biosciencedbc.jp/en/data-use]. Data users must comply with the NBDC guidelines [https://humandbs.biosciencedbc.jp/en/guidelines]. Access is granted for the entire period specified in the users' applications. NBDC administrative staff will respond to requests within one week. Publicly available RNA sequencing data of *KMT2A*-r infant B-ALL, RNA sequencing data of normal B-cell progenitors, and methylation array data of normal B-cell progenitors were downloaded from EGAS00001000246, GSE122982 and GSE45459, respectively. Single-cell RNA sequencing data of fetal liver cells were downloaded from the Development Cell Atlas Web Portal [https://developmentcellatlas.ncl.ac.uk/datasets/hca_liver/data_share/].
The following databases were used as described in "Methods": NCBI dbSNP [https://www.ncbi.nlm.nih.gov/snp/], Human Genetic Variation Database [https://www.hgvd.genome.med.kyoto-u.ac.jp] and Ensembl Biomart [https://feb2014.archive.ensembl.org/biomart]. The remaining data are available within the Article, Supplementary Information or Source Data file. Source data are provided with this paper.

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

## Acknowledgements

We thank M. Matsumura, N. Hoshino, K. Yin, F. Saito, and S. Shikata for their excellent technical assistance. We are also grateful to the FACS Core at the Institute of Medical Science, The University of Tokyo. This work was supported by the Japan Society of Promotion of Science (JSPS) KAKENHI (nos. JP17H04224, JP18K19467, JP19J11112, JP20H00528 and JP21K19405 to J.T.; no. JP19H05656 to S.O.; no. JP19H03685 to S.G.; and no. JP15K19601 to Y. Aoki); the Japan Agency for Medical Research and Development (AMED) (nos. JP15cm0106056h0005, JP19cm0106501h0004, JP16ck0106073h0003, JP19ck0106250h0003 and JP21cm0106501h0006 to S.O.); AMED Project for Cancer Research and Therapeutic Evolution (P-CREATE) (no. JP20cm0106509h9905 to J.T.; and nos. JP15cm0106139s0202 and JP20cm0106572h0001 to M. Takagi) and Practical Research for Innovative Cancer Control (no. JP20ck0106468h0002 to J.T.); AMED Project for Promotion of Cancer Research and Therapeutic Evolution (P-PROMOTE) (no. JP22ama221505h0001 to J.T.); Grants from the Ministry of Education, Culture, Sports, Science and Technology of Japan ("Program for Promoting Researches on the Supercomputer Fugaku" (Unraveling origin of cancer and diversity by large-scale data analysis and artificial intelligence technology, JPMXP1020200102)) (we used computational resources of supercomputer Fugaku provided by the RIKEN Center for Computational Science (Project ID: hp200138); the High Performance Computing Infrastructure System Research Project (nos. hp160219, hp170227, hp180198 and hp190158 to S.O.) (this research used computational resources of the K computer provided by the RIKEN Advanced Institute for Computational Science through the HPCI System Research project); Princess Takamatsu Cancer Research Fund to J.T.; and the Japan Foundation for Pediatric Research (no. 18-001 to T. Isobe). T. Isobe was supported by the Funai Foundation for Information Technology.

## Author contributions

T. Isobe, A.S.-O., A.N., K.Y., Y.N., H.U., K.W., M. Sekiguchi, M. Seki, S.K., A.K., T. Inaba, T. Morio, and S.O. performed sequencing experiments. T. Isobe, A.S.-O., and A.N. performed sequencing data analyses. T. Isobe, X.W., Y. Shiozawa, Y. Shiraishi, K.C., H.T., N.K.W., B.G., and S. Miyano developed bioinformatics pipelines. T. Isobe, M. Takagi, A.S.-O., C.Y., G.N., S.F., K.T., S.T., and H.A. performed methylation array analysis. T. Isobe, A.S.-O., and A.Y. performed ChIP experiments. T. Isobe, A.N., M. Tamura, and Y.T. performed FACS analyses. T. Isobe, S.A., R.T., A.T., R.A., I.K., T. Mikami, T.K., and S.G. performed functional assays. T. Isobe, M. Takagi, A.S.-O., and J.T. interpreted the results. R.S.K. and M.N.C. established cell lines. M. Takagi, M.H., M.K., F.I., M.E., T.D., N.K., Y. Arakawa, K.K., Y. Aoki, T. Ishihara, D.T., T. Miyamura, E.I., and S. Mizutani collected specimens. T. Isobe and A.S.-O. generated figures and tables. T. Isobe, M. Takagi, and J.T. wrote the manuscript. M. Takagi and J.T. coled the entire project. All authors participated in discussions and interpretation of the data and results.

## Competing interests

The authors declare no competing interests.

## Additional information

[1]Department of Pediatrics, Graduate School of Medicine, The University of Tokyo, Tokyo, Japan. [2]Division of Molecular Oncology, Department of Computational Biology and Medical Sciences, Graduate School of Frontier Sciences, The University of Tokyo, Tokyo, Japan. [3]Department of Hematology, Wellcome-MRC Cambridge Stem Cell Institute, University of Cambridge, Cambridge CB2 0AW, UK. [4]Department of Pediatrics and Developmental Biology, Tokyo Medical and Dental University, Tokyo, Japan. [5]Genome Science and Medicine Laboratory, Research Center for Advanced Science and Technology, The University of Tokyo, Tokyo, Japan. [6]Division of Cellular Therapy, The Institute of Medical Science, The University of Tokyo, Tokyo, Japan. [7]The Institute of Laboratory Animals, Tokyo Women's Medical University, Tokyo, Japan. [8]Department of Pathology and Tumor Biology, Graduate School of Medicine, Kyoto University, Kyoto, Japan. [9]Department of Pediatrics, Graduate School of Medicine, Kyoto University, Kyoto, Japan. [10]Department of Pediatrics, Hiroshima University Graduate School of Biomedical Sciences, Hiroshima, Japan. [11]Department of Pediatrics, School of Medicine, Teikyo University, Tokyo, Japan. [12]Department of Computational Biology and Medical Sciences, Graduate School of Frontier Sciences, The University of Tokyo, Kashiwa, Japan. [13]Department of Molecular Oncology and Leukemia Program Project, Research Institute for Radiation Biology and Medicine, Hiroshima University, Hiroshima, Japan. [14]Division of Genome Analysis Platform Development, National Cancer Center Research Institute, Tokyo, Japan. [15]Department of Integrated Data Science, M&D Data Science Center, Tokyo Medical and Dental University, Tokyo, Japan. [16]Leukaemia Translational Research Laboratory, Telethon Kids Cancer Centre, Telethon Kids Institute, University of Western Australia, Perth, WA, Australia. [17]Department of Clinical Haematology, Oncology, Blood and Marrow Transplantation, Perth Children's Hospital, Perth, WA, Australia. [18]Curtin Medical School, Curtin University, Perth, WA, Australia. [19]School of Biomedical Sciences, University of Western Australia, Nedlands, WA, Australia. [20]Laboratory for Human Disease Models, RIKEN Center for Integrative Medical Sciences, Yokohama, Japan. [21]Department of Pediatrics, Ehime University Graduate School of Medicine, Ehime, Japan. [22]Division of Cancer Immunodiagnostics, Children's Cancer Center, National Center for Child Health and Development, Tokyo, Japan. [23]Department of Pediatrics, Graduate School of Medicine, Mie University, Tsu, Japan. [24]Department of Pediatric Hematology and Oncology Research, National Research Institute for Child Health and Development, Tokyo, Japan. [25]Department of Hematology/Oncology, Saitama Children's Medical Center, Saitama, Japan. [26]Department of Pediatric Oncology, National Cancer Center Japan, Tokyo, Japan. [27]Department of Pediatrics, Nara Medical University, Kashihara, Japan. [28]Division of Leukemia and Lymphoma, Children's Cancer Center, National Center for Child Health and Development, Tokyo, Japan. [29]Department of Pediatrics, Osaka University Graduate School of Medicine, Osaka, Japan. [30]Tsuruoka Metabolomics Laboratory, National Cancer Center, Tsuruoka, Japan. [31]Institute for the Advanced Study of Human Biology (WPI-ASHBi), Kyoto University, Kyoto, Japan. [32]Department of Molecular Hematology, Karolinska Institute, Stockholm, Sweden. [33]These authors contributed equally: Tomoya Isobe, Masatoshi Takagi, Aiko Sato-Otsubo. ✉e-mail: m.takagi.ped@tmd.ac.jp; jtakita@kuhp.kyoto-u.ac.jp

