## [Peer Review File · Nature Communications]

Multi-omics analysis defines highly refractory RAS burdened immature subgroup of infant acute lymphoblastic leukemiaReviewers' Comments:

Reviewer #1:

Remarks to the Author:

The manuscript by Takita and colleagues provides an integrative genomics-based approach to subtyping of KMT2A-rearranged acute lymphoblastic leukemias in infants. The authors present a set of results that produces five distinct subtypes of the disease based on dual-omic analysis of transcription and methylation. The resulting subtypes are highly correlated with disease outcome based on Kaplan-Meier analysis, and should provide a baseline for further genomics-based risk stratification. In addition, the use of novel deep sequencing approaches was able to identify sub-clonal RAS pathway gene mutation discovery, indicating that patients with the highest mutational burden in the RAS pathway have the worst outcomes and may be appropriate to consider treatment with drugs targeting this pathway. Taken together, the manuscript is well-written and conveys a logical and strategic approach to the results obtained and the conclusions/next steps. As such, the significance of this study to the field of pediatric (infant) ALL risk stratification and treatment is high.

Although the work is well presented, there are minor comments that should be addressed to clarify the presentation of data and methods, as follows:

Abstract; final sentence 'provide a rationale for the future development of genomics-guided individualized therapy' but what about risk stratification by genomics? The authors may wish to modify, since this reflects text in the conclusion of the paper.

Samples: What were the treatment differences between the patients studied? A table of information about each patient and therapeutic modalities, duration of treatment, and other pertinent clinical data should be included. Also need to include their IC subtype assignment in this table. Consider adding to Table 1 in supplementary data.

Results: line 113 through 115 is written in a very confusing way, by pointing out 'splice variants that skip exon 11 of KMT2A'...are the authors calling the KMT2A gene fusions "splice variants"? Certainly on one hand they are, but this is a very unorthodox way of describing a gene fusion event, especially because splice variants cause exon skipping in intragenic deletion events, which are not gene fusions. It would seem that, for the sake of clarity, the sentence should be re-written and refer to gene fusions, not to splice site mutations that lead to exon skipping.

Line 146: The description here and the Figure 1a data display do not correlate. The description states the three HOXA- subtypes are clearly separated by fusion partners whereas there are, for example four different fusion partners delineated in Fig 1a as being IC5 subtype. The description in line 116 should be more precise, according to the data in Fig 1a. This is restated on line 350-351 and should also be corrected.

Line 201: was WES performed for only 19/61 and 12/23 cases due to lack of a comparator normal, or due to limited amounts of tumor DNA or both? What was the comparator normal tissue used for WES? While the sub-clonal detection of additional RAS/MAPK pathway genes is certainly of interest, can the authors speculate on how the sub-clonal mutations are impacting disease outcomes? This seems to be difficult to imagine, since they are only present in small proportions of tumor cells based on VAF.

Line 472: please provide details on the number of known KMT2A fusions that were missed by Genomon analysis, also the number predicted by genomic SV predictions (and which algorithm was used for SV prediction) and proven correct should be provided.

Line 567: Were the deep-seq libraries made with universal molecular identifiers (UMI) included on the library adapters? This is standard practice for high depth sequencing so that polymerase errors are not being reported as true variants. Without an orthogonal evaluation or verification of the low VAF

variants from deep-seq, having UMI-corrected variant identification is critically important.

Reviewer #2:

Remarks to the Author:

In the current study, Isobe and colleagues performed a comprehensive multi-omics unsupervised clustering of infant ALL with KMT2A-rearrangement. A dual-omics clustering of expression and methylation data identified previously unknown subgroups within the IRX and HOXA subtypes. One interesting finding is that the IRX subtype can be classified into two groups. One of them, Integrative Cluster 2 (IC2), have particularly poor survival. The IC2 subgroup was also found to have high-frequency RTK-RAS mutations. Further functional studies confirmed that KMT2A fusion is capable of inducing either IRX or HOXA subtypes, and the subtype selection between IRX and HOXA is potentially driven by cell origin. Overall, this is a solid study that provides many new insights into the molecular subtyping and mechanisms of infant ALL. The findings are novel and clinically relevant. There are some questions that might be better addressed.

Major:

1. Based on figure 1a, it seems IC2 patients might also be associated with younger age?
2. One major conclusion is that the IC2 subgroup is associated with a higher frequency of RTK-RAS mutations than other subgroups. Is it because the IC2 cases have higher mutation burdens, or more heterogeneous clonal composition, or neither (which would indicate the mutations are solely enriched in RTK-RAS pathway genes)? Also do the mutations in IC2 cases have higher VAFs than those in other subgroups?
3. It has been previously reported that RTK-RAS mutations in infant ALL are mostly subclonal and usually disappear in relapse (<https://www.ncbi.nlm.nih.gov/pubmed/25730765>). Would that contradict the current finding that RTK-RAS mutations are the most pronounced genomic feature of IC2 and potential driver of the poor prognosis?
4. In figure 3a, one interesting observation is that FLT3 and KRAS/NRAS are mutually exclusive in most subgroups except for IC2. Could the co-occurrence of FLT3 and RAS mutations be a possible explanation of the poorer prognosis of IC2 in these patients? Further, based on the current data, is it possible to infer whether the FLT3 and KRAS/NRAS are from the same or different subclones in the IC2 cases with both of them?

Minor:

1. Figure 3a, since the genomic characterization was performed using different platforms (WES in some cases and deep sequencing in all), it would be helpful to add a horizontal bar to indicate the sequencing platforms used for each case. This information would be useful to rule out potential bias due to technical sensitivity.
2. Line 115 "accounting for 95% of the total fusions identified" is confusing. Does it mean that 105/111 fusions are KMT2A exon 11 skipping?
3. In the discussion "the number of RTK-RAS mutations, but not simple positivity, significantly predicts patient prognosis", it might be pre-mature to claim the "number of RTK-RAS mutations" as the predictive biomarker, without knowing the exact cause of the observed high numbers of RTK-RAS mutations. Again, this is related to the previous question.

Reviewer #3:

Remarks to the Author:

Isobe et al use a combination of RNASeq and DNA methylation to define 5 clusters within infant KMT2Ar infant ALL, a very difficult to treat leukemia with poor outcomes. Previously, transcriptome analysis has defined a group based on IRX overexpression and a mutually exclusive HOXA group. In this manuscript, the authors breakdown the IRX group into two separate clusters, with IC2 being associated with particularly poor prognosis. The authors show that the IRX and HOXA subtypes are mostly mutually exclusive. Using single cell sequencing they focus on one of the "double" cases to show that the IRX and HOXA cells are not the same cells, though there are a few double cells. In general IRX subtypes tend to go along with leukemia that is earlier in development, with expression of KMT2A-AFF1 in HSC/MPP leading to IRX subtype where as expression of this construct in later, lymphoid primed progenitors, led to a HOXA subtype. The authors also used deep sequencing to discover a very high prevalence of RAS pathway and FLT3 mutations, many of these present as small clones.

This paper is certainly informative and adds to the literature of this very aggressive leukemia. The idea of the "molecular switch" guided by the KMT2A-AFF1 fusion in early progenitors is interesting and will be great to explore more in the future.

I do have a few questions/clarifying points:

1. Cohort: The authors used samples from 61 leukemias initially and then have a another cohort of 23 samples which are labeled as "validation" in Table 1 but also referred to as an "extended cohort" in extended figure 3. Given the small numbers, I am not sure that the extended cohort analysis adds to the paper—for example, the survival analysis for this smaller cohort is not significantly different between the clusters. Can these 23 samples be combined with the initial cohort and perhaps that would improve the outcome analysis Figure 1? Can published MLLr ALL datasets be used for RNASeq validation?
2. Some infant ALL studies have used other clinical factors as predictors of poor outcome—high WBC, steroid response, young age. How are these other factors (especially steroid response) correlate with the clustering?
3. The finding of very small RAS and FLT3 clones is very interesting. Given the very small size of the clones, how does their presence affect the leukemia biology? Do the authors have relapse samples from some of these patients? Do the RAS clones go up in size at the time of relapse?
4. The presence of RAS/FLT3 mutations was not associated with EFS/OS in this study, but the presence of multiple RAS clones was associated with a worse EFS and this seems to be more present in the IC2 cluster. IF the authors focus on samples with the higher VAF, for example >25%, is there a EFS/OS difference then?
5. The authors perform network analysis for IRX1. How does it compare to IRX2? Seems both are highly correlated in the RNASeq analysis.
6. Survival analysis of KMT2A-AFF1 fusions in extended figure 1e is not statistically significant and thus should not be stated as "poorer" in the text (lines 131-133).
7. Infant KMT2Ar leukemia often has myeloid co-expression and can undergo lineage switch to myeloid disease at the time to relapse. How are myeloid marker genes expression between the ICs? Would one cluster be more likely to undergo a myeloid switch versus another?

8. The authors express KMT2A-AFF1 in embryonic cells from different stages of development to show that expression in HSC or MPP leads to a IRX1 type of leukemia vs expression in a lymphoid progenitor to a HOX subtype. How would expression of one of the other KMT2A fusions compare?

9. I am interested in this "Molecular switch" between IRX/HOX subtypes. Perhaps beyond the scope of this paper, would be interesting to see if this switch in subtypes happens with relapse.

10. There is only one supplementary table included. I could not locate Supplementary Table 2 and later.

Reviewer #4:

Remarks to the Author:

In this manuscript, the authors have performed RNA sequencing, methylation array analysis, whole exome and targeted deep sequencing on 84 infants with KMT2A-rearranged leukemia, using multi-omics to define five robust integrative clusters (ICs) in KMT2A-rearranged ALL. What's more, they revealed that the number of RAS pathway mutations predicts prognosis and plays an important way in IC2. They provided a more detailed ALL subtypes, which will be beneficial to genomics-guided individualized therapy, and those multi-omics data could be useful ALL resources database. Taken together, I think this manuscript is probably publishable, but major revision is needed.

Major:

1. The cohort only have ALL samples, but the normal sample should be included too, especially for transcriptome, DNA methylome and single-cell data. Though IRX-subtype and HOXA-subtype were identified by RNA-seq, some features of subtype may be shared by normal samples. Using control cohort helps define clearer ALL subtypes. What's more, When the authors validate this clustering results, they should include normal cohort too.

2. In Fig.1a, expression heatmap and methylation heatmap looks alike, whether the methylation probes overlapped with expression heatmap genes? If so, can single omics (transcriptome or DNA methylome) do this clusters?

3. The authors do many omics, such as transcriptome, DNA methylome and WES, but the analysis of each omics is separate, and the authors can try a joint analysis to get more novelty.

4. little investigation of the molecular heterogeneity of KMT2A-r infant ALL in DNA methylation. The authors identified more refined subtypes based on both transcriptome and DNA methylation, which indicated that the DNA methylation might be an important determinant of molecular profiles in KMT2A-r infant ALL. Thus, they need to explore the DNA methylation landscape more deeply and detailedly.

Minor:

1. For Fig2g, Endothelial signature gene set is small number, and it hard to convincing. It's suggested for authors to collect better Endothelial signature gene set and re-analyze it.

2. The authors have showed that enrichment analysis of hypo-methylated regions in HOXA subtype and IRX subtype, how about hyper-methylated regions enrichment analysis?

3. The authors successfully identified five subgroups in KMT2A-r infant ALL by both transcriptome and DNA methylome-based clustering, while only two subgroups identified by transcriptome-based clustering. Their results demonstrated that multi-omics analysis performed better than single-omics clustering in revealing more complex and clinically relevant disease subtypes. The reason should be given in the discussion section.

We would like to thank the reviewers for their constructive comments, which helped us to further improve our manuscript. We have conducted additional data analysis and revised parts of our manuscript where appropriate. We have provided a point-by-point response below in blue font and marked all changes in the manuscript text in red font.

Reviewer #1, expertise in paediatric precision oncology/genomics (Remarks to the Author):

The manuscript by Takita and colleagues provides an integrative genomics-based approach to subtyping of *KMT2A*-rearranged acute lymphoblastic leukemias in infants. The authors present a set of results that produces five distinct subtypes of the disease based on dual-omic analysis of transcription and methylation. The resulting subtypes are highly correlated with disease outcome based on Kaplan-Meier analysis, and should provide a baseline for further genomics-based risk stratification. In addition, the use of novel deep sequencing approaches was able to identify sub-clonal RAS pathway gene mutation discovery, indicating that patients with the highest mutational burden in the RAS pathway have the worst outcomes and may be appropriate to consider treatment with drugs targeting this pathway. Taken together, the manuscript is well-written and conveys a logical and strategic approach to the results obtained and the conclusions/next steps. As such, the significance of this study to the field of pediatric (infant) ALL risk stratification and treatment is high.

Although the work is well presented, there are minor comments that should be addressed to clarify the presentation of data and methods, as follows:

Abstract; final sentence 'provide a rationale for the future development of genomics-guided individualized therapy' but what about risk stratification by genomics? The authors may wish to modify, since this reflects text in the conclusion of the paper.

We thank the reviewer for this suggestion. As the reviewer suggests, genomics-based risk stratification is one of the major implications of our study. We have therefore revised the last sentence of the abstract as follows:

“Our findings highlight the previously under-appreciated intra- and inter-patient heterogeneity of *KMT2A*-rearranged infant ALL and provide a rationale for the future development of genomics-guided risk stratification and individualized therapy.” (Lines 74-77)

Samples: What were the treatment differences between the patients studied? A table of information about each patient and therapeutic modalities, duration of treatment, and other pertinent clinical data should be included. Also need to include their IC subtype assignment in this table. Consider adding to Table 1 in supplementary data.

In response to the reviewer's comment, we have now added relevant clinical data as well as the IC subtype of each patient in Supplementary Table 1 (Columns I-Q and S-T). Further details of the treatment protocols are separately reported in the respective clinical trial papers cited in the Methods section (line 525).

Results: line 113 through 115 is written in a very confusing way, by pointing out 'splice variants that skip exon 11 of *KMT2A*'...are the authors calling the *KMT2A* gene fusions "splice variants"? Certainly on one hand they are, but this is a very unorthodox way of describing a gene fusion event, especially because splice variants cause exon skipping in intragenic deletion events, which are not gene fusions. It would seem that, for the sake of clarity, the sentence should be re-written and refer to gene fusions, not to splice site mutations that lead to exon skipping.

We apologize for the confusing wording. We have observed alternative splicing of *KMT2A* fusions skipping the exon 11 of *KMT2A*, resulting in the identification of two different *KMT2A* fusion transcripts within a single patient sample in 25 of 61 cases (e.g., *KMT2A* exon10-*AFF1* and *KMT2A* exon11-*AFF1* in MLL_10_004; Supplementary Table 3). This skipping of exon 11 has been reported in normal and fused *KMT2A* gene (Meyer et al., PMID: 15626757). To make this clearer, we have revised the text to:

"In 25 of 61 cases (41%), alternative splicing of *KMT2A* fusions skipping the exon 11 of *KMT2A* was detected." (Lines 117-118)

Line 146: The description here and the Figure 1a data display do not correlate. The description states the three HOXA- subtypes are clearly separated by fusion partners whereas there are, for example four different fusion partners delineated in Fig 1a as being IC5 subtype. The description in line 116 should be more precise, according to the data in Fig 1a. This is restated on line 350-351 and should also be corrected.

We thank the reviewer for raising this point. We agree that the separation of fusion partners in the three HOXA-type ICs is not completely exclusive. To quantitatively describe our observation, we statistically evaluated the association between fusion partners and the IC assignment and have revised our statements:

"Thereby, the IRX-subtype was divided into IC1 and IC2, whereas the HOXA-subtype was split into three ICs, which significantly correlated with fusion partners: *MLL1* (IC3; Fisher's exact $P = 3.7 \times 10^{-8}$), *MLL3* (IC4; Fisher's exact $P = 5.6 \times 10^{-6}$), and *AFF1* (IC5; Fisher's exact $P = 5.7 \times 10^{-6}$), respectively." (Lines 147-150)

"Notably, our unsupervised analysis revealed significant correlations between multi-omics molecular profiles and fusion partners, particularly in the HOXA-subtype: *MLL1* (IC3), *MLL3* (IC4) and *AFF1* (IC5)." (Lines 405-407)

Line 201: was WES performed for only 19/61 and 12/23 cases due to lack of a comparator normal, or due to limited amounts of tumor DNA or both? What was the comparator normal tissue used for WES? While the sub-clonal detection of additional RAS/MAPK pathway genes is certainly of interest, can the authors speculate on how the sub-clonal mutations are impacting disease outcomes? This seems to be difficult to imagine, since they are only present in small proportions of tumor cells based on VAF.

We thank the reviewer for raising this important point. Firstly, WES was performed in the cases for which paired normal samples were available. For the paired normal samples, we used normal CD3+ T-cells sorted and expanded from the diagnostic samples. We apologize for not making this clear in our original submission. We have added the following details in the Methods section (lines 614-617):

“WES was performed in the cases for which paired tumor and normal DNA samples were available. For the paired normal samples, CD3+ T-cells were sorted from the diagnostic samples using MACS beads (Miltenyi). Isolated CD3+ T-cells were expanded using T Cell Activation/Expansion Kit (Miltenyi), and DNA was extracted using QIAamp DNA minikit (QIAGEN).”

Regarding the prognostic significance of subclonal RAS pathway mutations, Ma and colleagues recently analyzed 20 pairs of diagnostic and relapse pediatric B-ALL samples (PMID: 25790293) and showed that relapse founder clones are often subclonal at diagnosis, with a median population frequency of 7%. They have further revealed that subclonal RAS pathway mutations with VAFs at diagnosis of ~2% can expand and initiate relapse. Therefore, we speculate that the existence of multiple RTK-RAS mutant subclones in IC2 may increase the likelihood that at least one of these mutant subclones will survive treatment and lead to relapse. We have now added this speculation to the text in the Discussion section (lines 364-373):

“Recently, Ma and colleagues have shown that the relapse founder clone in pediatric B-ALL often originates from a minor subclone at diagnosis, and that subclonal RAS pathway mutations with low VAFs of ~2% can expand and seed relapse³². Therefore, the existence of multiple different RTK-RAS mutant subclones in IC2 may increase the chances that at least one of these mutant subclones will break through treatment and lead to relapse. In fact, our sequencing of a paired relapse sample from a case of IC2 (UT_INF_001) demonstrated that two RTK-RAS mutations at diagnosis had contributed to relapse in this case. However, disappearance of RTK-RAS mutations at relapse has also been reported⁸, indicating that RTK-RAS wild-type subclones can compete and predominate in such cases, necessitating further investigation into the mechanisms of how RTK-RAS mutant subclones contribute to relapse.”

Line 472: please provide details on the number of known KMT2A fusions that were missed by Genomon analysis, also the number predicted by genomic SV predictions (and which algorithm was used for SV prediction) and proven correct should be provided.

Thank you for making this point. SVs were detected using Genomon as described in the Methods section (lines 592-593), and the “Pipeline” column of Supplementary Table 3 contains the information about which fusions were identified with which pipeline. To clarify this, we have added the following sentence in the Methods section (lines 546-548):

“Overall, 85 fusions were identified by Genomon, and 21 and 5 fusions were restored by Pizzly and SV-based prediction, respectively (Supplementary Table 3).”

Line 567: Were the deep-seq libraries made with universal molecular identifiers (UMI) included on the library adapters? This is standard practice for high depth sequencing so that polymerase errors are not being reported as true variants. Without an orthogonal evaluation or verification of the low VAF variants from deep-seq, having UMI-corrected variant identification is critically important.

We agree with the reviewer that it is important to use UMI-incorporated sequencing or perform orthogonal validation for the detection of low-VAF variants. Since our deep-seq does not include UMIs, we have performed amplicon-deep sequencing to validate all low-VAF variants with VAFs ≤ 0.10 . In total, 50 variants were subjected to the orthogonal DNA sequencing, and all 50 variants were successfully validated. A Reviewer Table containing the results of validation sequencing has been uploaded separately as a spreadsheet file. Accordingly, we have revised the Methods section in lines 634-640:

“Among the variants identified with deep-seq, all subclonal SNVs/indels with VAFs ≤ 0.10 were subjected to PCR-based amplicon deep sequencing for validation. For PCR amplification, a NotI restriction site was attached to each primer as a linker sequence. Amplified products were digested with NotI, ligated, fragmented, and then used for deep sequencing library preparation as previously described⁵⁴. In total, 90 of 96 (94%) WES-based candidate mutations and 50 of 50 (100%) deep-seq-based subclonal mutations were validated across the discovery and extended cohorts.”

Reviewer #2, expertise in multi-omics and bioinformatics (Remarks to the Author):

In the current study, Isobe and colleagues performed a comprehensive multi-omics unsupervised clustering of infant ALL with KMT2A-rearrangement. A dual-omics clustering of expression and methylation data identified previously unknown subgroups within the IRX and HOXA subtypes. One interesting finding is that the IRX subtype can be classified into two groups. One of them, Integrative Cluster 2 (IC2), have particularly poor survival. The IC2 subgroup was also found to have high-frequency RTK-RAS mutations. Further functional studies confirmed that KMT2A fusion is capable of inducing either IRX or HOXA subtypes, and the subtype selection between IRX and HOXA is potentially driven by cell origin. Overall, this is a solid study that provides many new insights into the molecular subtyping and mechanisms of infant ALL. The findings are novel and clinically relevant. There are some questions that might be better addressed.

Major:

1. Based on figure 1a, it seems IC2 patients might also be associated with younger age?

In response to the reviewer's comment, we performed a statistical comparison of clinical characteristics including age between the five ICs and found no significant difference. This has been added as Supplementary Table 4 and the following statement has been added to the text (lines 150-152):

“Other patient characteristics, including known clinical prognostic factors such as age, did not correlate significantly with the IC assignment (Supplementary Table 4).”

2. One major conclusion is that the IC2 subgroup is associated with a higher frequency of RTK-RAS mutations than other subgroups. Is it because the IC2 cases have higher mutation burdens, or more heterogeneous clonal composition, or neither (which would indicate the mutations are solely enriched in RTK-RAS pathway genes)? Also do the mutations in IC2 cases have higher VAFs than those in other subgroups?

We thank the reviewer for raising this important point. We have now compared the exome-wide mutation burdens in our discovery cases studied with WES (n=19). In these patients, the total number of mutations per exome did not differ significantly between ICs (please see Reviewer Fig. 1a,b below). However, whether the mutations are solely enriched in the RTK-RAS pathway cannot be concluded, because when comparing only the WES-based mutations, even RTK-RAS pathway mutations are not significantly different (Reviewer Fig. 1c), due to the limited sample size assessed with WES as well as the lower sensitivity of WES for subclonal mutations. Since we took advantage of deep-sequencing by targeting a panel of recurrently mutated genes in infant ALL, another large-scale study utilizing exome-wide deep-sequencing would be needed to compare mutation rates within and outside the RTK-RAS pathway.

Furthermore, VAFs of RTK-RAS mutations were similarly low with medians of <0.2 in all five ICs (Reviewer Fig. 1d), suggesting that the individual VAFs may not be a crucial genomic feature to distinguish the ICs. Please see comments 3 and 4 below for further discussion on the clonal architecture.

Reviewer Fig. 1

Reviewer Fig. 1. Comparison of the numbers and VAFs of mutations in the ICs. **a-c**, The numbers of all mutations (a), non-RTK-RAS mutations (b) and RTK-RAS mutations (c) per case in the discovery cases examined with WES (n = 19). **d**, VAFs of all individual mutations in the RTK-RAS pathway identified with deep-seq.

3. It has been previously reported that RTK-RAS mutations in infant ALL are mostly subclonal and usually disappear in relapse (<https://www.ncbi.nlm.nih.gov/pubmed/25730765>). Would that contradict the current finding that RTK-RAS mutations are the most pronounced genomic feature of IC2 and potential driver of the poor prognosis?

Again, we thank the reviewer for this important point. As the reviewer points out, the clinical and biological significance of subclonal RTK-RAS mutations is still under debate, as disappearance of RTK-RAS mutations at relapse has been reported. In the paper suggested by the reviewer (PMID: 25730765), Andersson and colleagues examined five relapse samples whose paired diagnostic samples had one or two mutations in the RTK-RAS pathway. They identified two cases with maintained and expanded RTK-RAS mutant clones at relapse as well as one case with a persisting RTK-RAS mutant subclone, while in the remaining two cases the RAS mutant subclones were eradicated.

To our knowledge, Ma and colleagues have conducted another comprehensive study on the subclonal architecture and evolution from diagnosis to relapse of pediatric B-ALL using deep sequencing (PMID: 25790293). Although they did not include *KMT2A-r* ALL, they showed that nine out of 20 B-ALL cases had multiple subclonal mutations in the RTK-RAS pathway genes at diagnosis. More importantly, they reported that in seven of the nine cases, the multiple RTK-RAS mutations converged so that only one mutation (*NRAS* in five cases and *KRAS* in two cases) expanded and the others disappeared at relapse. In the remaining two out of the nine cases, two different RTK-RAS mutations persisted from diagnosis to relapse.

These complex observations may indicate the following:

- 1) Since RTK-RAS pathway mutations are mostly subclonal, they may not be necessary for disease initiation or maintenance.
- 2) It is also possible that RTK-RAS wild-type subclones compete with and predominate over RTK-RAS mutant subclones to initiate relapse, as was observed in the two out of five cases in the Andersson cohort.
- 3) However, RTK-RAS mutant subclones may have a greater advantage in overcoming treatment, since RTK-RAS mutant subclones contributed to relapse in nine out of nine and three out of five cases in the Ma and Andersson cohorts, respectively.
- 4) Therefore, the existence of multiple different RTK-RAS mutant subclones in IC2 may increase the chances that at least one of these mutant subclones will break through treatment and lead to relapse.

We have included these speculations in the Discussion section (lines 364-373) as a possible interpretation of our results. Concurrently, we have also toned down our statement about the predictive value of the number of RTK-RAS mutations (lines 380-383), since further validation of the prognostic significance and mechanistic studies are still needed in the future.

4. In figure 3a, one interesting observation is that *FLT3* and *KRAS/NRAS* are mutually exclusive in most subgroups except for IC2. Could the co-occurrence of *FLT3* and *RAS* mutations be a possible explanation of the poorer prognosis of IC2 in these patients? Further, based on the current data, is it possible to infer whether the *FLT3* and *KRAS/NRAS* are from the same or different subclones in the IC2 cases with both of them?

We thank the reviewer for this interesting suggestion. We performed a survival analysis comparing patients with and without concurrent *FLT3* and *RAS* (*KRAS/NRAS*) mutations but

found no significant difference in survival. We have added this result as Supplementary Figure 6d and revised the text accordingly (lines 255-257):

“Of note, although neither simple positivity of RTK-RAS pathway mutations nor co-occurrence of *FLT3* and *RAS* mutations was significantly associated with survival rates (Supplementary Fig. 6a-d)”

We also agree with the reviewer that it would be of particular interest to know whether the *FLT3* and *RAS* mutations are in the same or different subclones. Given the mutually exclusive pattern in other ICs, we speculate that they occur in different subclones. However, determining the clonal architecture would require either samples at different time points or single-cell-based approaches. Chronological analysis (e.g., paired sequencing of diagnostic and relapse samples) would reveal co-existing mutations by subgrouping mutations with a similar increase or decrease in VAFs over time, as shown in Ma and colleagues’ study discussed in comment 3 above (PMID: 25790293). Otherwise, single-cell DNA sequencing would also elucidate the co-occurrence of mutations even with only diagnostic samples. Although we would be interested in pursuing this in the future, we feel that it is beyond the scope of this paper.

Minor:

1. Figure 3a, since the genomic characterization was performed using different platforms (WES in some cases and deep sequencing in all), it would be helpful to add a horizontal bar to indicate the sequencing platforms used for each case. This information would be useful to rule out potential bias due to technical sensitivity.

Following the reviewer’s suggestion, we have now added a horizontal bar in the figure (revised Figure 4a) to show the sequencing platforms used in each case.

2. Line 115 “accounting for 95% of the total fusions identified” is confusing. Does it mean that 105/111 fusions are *KMT2A* exon 11 skipping?

We apologize for the confusing description. What we have observed is that 106/111 fusions are involving *KMT2A* gene, and among these, 25 *KMT2A* fusion transcripts that skip exon 11 are included. We have revised the text in lines 115-118:

“In total, 111 fusion transcripts were identified and experimentally validated (Supplementary Table 3), of which 95% were *KMT2A*-related fusions. In 25 of 61 cases (41%), alternative splicing of *KMT2A* fusions skipping the exon 11 of *KMT2A* was detected.”

3. In the discussion “the number of RTK-RAS mutations, but not simple positivity, significantly predicts patient prognosis”, it might be pre-mature to claim the “number of RTK-RAS mutations” as the predictive biomarker, without knowing the exact cause of the observed high numbers of RTK-RAS mutations. Again, this is related to the previous question.

As we discussed in our response to comment 3 above, we agree with the reviewer that it would be premature to draw conclusions before we and other groups can confirm the prognostic significance of these subclonal mutations using a cohort including paired diagnostic and relapse samples. We have toned down our statement in lines 380-383:

“Because infant ALL is particularly enriched with subclonal RTK-RAS mutations and our results show potential contribution of subclonal diversity to higher relapse rates, accurate recognition of subclonal RTK-RAS mutations based on deep sequencing would be of particular importance for clinical decision making.”

Reviewer #3, expertise in acute leukemia genomics and models (Remarks to the Author):

Isobe et al use a combination of RNASeq and DNA methylation to define 5 clusters within infant KMT2Ar infant ALL, a very difficult to treat leukemia with poor outcomes. Previously, transcriptome analysis has defined a group based on IRX overexpression and a mutually exclusive HOXA group. In this manuscript, the authors breakdown the IRX group into two separate clusters, with IC2 being associated with particularly poor prognosis. The authors show that the IRX and HOXA subtypes are mostly mutually exclusive. Using single cell sequencing they focus on one of the “double” cases to show that the IRX and HOXA cells are not the same cells, though there are a few double cells. In general IRX subtypes tend to go along with leukemia that is earlier in development, with expression of KMT2A-AFF1 in HSC/MPP leading to IRX subtype where as expression of this construct in later, lymphoid primed progenitors, led to a HOXA subtype. The authors also used deep sequencing to discover a very high prevalence of RAS pathway and FLT3 mutations, many of these present as small clones.

This paper is certainly informative and adds to the literature of this very aggressive leukemia. The idea of the “molecular switch” guided by the KMT2A-AFF1 fusion in early progenitors is interesting and will be great to explore more in the future.

I do have a few questions/clarifying points:

1. Cohort: The authors used samples from 61 leukemias initially and then have a another cohort of 23 samples which are labeled as “validation” in Table 1 but also referred to as an “extended cohort” in extended figure 3. Given the small numbers, I am not sure that the extended cohort analysis adds to the paper—for example, the survival analysis for this smaller cohort is not significantly different between the clusters. Can these 23 samples be combined with the initial cohort and perhaps that would improve the outcome analysis Figure 1? Can published MLLr ALL datasets be used for RNASeq validation?

We thank the reviewer for raising this point, and we apologize for the inconsistent labelling of our secondary cohort. As suggested by the reviewer, we have performed a methylation-based re-clustering using the entire cohort of 84 infants. However, even with the 84 cases, methylation-based single-omics clustering did not identify more than two stable clusters, as clustering stability

based on average silhouette width drops rapidly with cluster numbers >2 (Reviewer Fig. 1a,b). This clustering ($k = 2$) also did not correlate with patient outcomes (Reviewer Fig. 1c).

Reviewer Fig. 1

Reviewer Fig. 1. Methylation-based clustering of the entire cohort of 84 infants. **a**, DNA methylation heatmap of the methylation-based clusters ($k = 2$). **b**, Average silhouette widths for $k = 2$ to $k = 8$. **c**, Survival analysis based on the methylation-based clusters of 84 infants.

On the other hand, although statistical significance was not achieved, the 23 cases of methylation cohort showed similar methylation profiles (Extended Data Fig. 3d), survival rates (Extended Data Fig. 3e), and mutational landscapes (Extended Data Fig. 3f) to our discovery cohort (Fig. 1a-c and Fig. 4a). Therefore, we consider that these 23 cases are still of some value and could be presented as a secondary cohort. However, as the validation power of this small secondary cohort is limited, we have amended the name of the cohort to “extended cohort” throughout the text and data.

Finally, we have performed additional expression-based validation using a published dataset by Andersson and colleagues of St Jude Children’s Research Hospital (PMID: 25730765; accession no. EGAS00001000246), which includes RNA sequencing data for 31 diagnostic samples of *KMT2A*-r infant B-ALL with typical partner genes (i.e., 16 *AFF1*, six *MLLT1*, five *MLLT3* and four *MLLT10* cases). We have employed the same KNN classifier modelling approach that we used for methylation-based validation in our extended cohort. As a result, the St Jude cohort also recapitulated the cluster-specific expression profiles as well as the characteristic separation of specific fusion partners in the HOXA-type ICs. Since whole genome sequencing was used for the exploratory purpose of their study, the number of RAS pathway mutations cannot be fully

evaluated and should be validated in the future using a deep sequencing method. However, all five IC2 cases had one or more RTK-RAS pathway mutations, which was a significantly higher frequency than the other ICs (100% vs 35%; Fisher's exact $P = 0.012$). These results have been added as Supplementary Fig. 8 and the following statement has been added to the text (lines 272-278):

"Finally, for additional validation, we exploited a published RNA sequencing dataset of 31 diagnostic samples of *KMT2A*-r infant B-ALL (EGAS00001000246)⁸. By building and applying an expression-based KNN classifier, the 31 infants were assigned with IC labels that recapitulated the cluster-specific expression patterns and characteristic distribution of fusion partners (Supplementary Fig. 8). Again, IC2 was shown to have a significantly higher frequency of RTK-RAS pathway mutations than the other ICs (100% vs. 35%; Fisher's exact $P = 0.012$), although the number of subclonal mutations should be further validated using a deep sequencing method."

The procedures for generating the expression-based classifier have also been added to the Methods section (lines 600-612).

2. Some infant ALL studies have used other clinical factors as predictors of poor outcome—high WBC, steroid response, young age. How are these other factors (especially steroid response) correlate with the clustering?

In response to the reviewer's comment, we compared clinical characteristics between the five ICs and found no significant difference, although data on steroid response are not fully available for most patients treated in the MLL96/98 trial. The relevant clinical data have been added to Supplementary Table 1 (Columns I-Q and S-T), and the results of the statistical comparison have been added as new Supplementary Table 4. Accordingly, the following statement has been added to the text (lines 150-152):

"Other patient characteristics, including known clinical prognostic factors such as age, did not correlate significantly with the IC assignment (Supplementary Table 4)."

3. The finding of very small RAS and FLT3 clones is very interesting. Given the very small size of the clones, how does their presence affect the leukemia biology? Do the authors have relapse samples from some of these patients? Do the RAS clones go up in size at the time of relapse?

We thank the reviewer for this important suggestion. Since no relapse samples were collected in our previous clinical trials, only one relapse sample (UT_INF_001_R) could be newly collected for the evaluation of this point. Since our original deep-seq of the diagnostic sample (UT_INF_001) had identified mutations in *FLT3*, *PTPN11* and *TP53*, we have performed PCR-based amplicon deep sequencing of these three mutations in the diagnostic and relapse samples. As a result, all three mutations were identified in the relapse sample with higher VAFs than in the diagnostic sample (please see our new Supplementary Table 14). This suggests the contribution of these mutations to relapse in this case, although the significance of RTK-RAS mutations as relapse drivers should be further evaluated with a larger number of paired diagnostic/relapse samples. Accordingly, we have added the following statement in the text (lines 261-265):

“Indeed, in one case of IC2 (UT_INF_001) for which a relapse sample was available, the two RTK-RAS mutations (*FLT3* and *PTPN11*) at diagnosis increased the VAFs at relapse (Supplementary Table 14), indicating the contribution of these mutations to relapse in this case, although the significance of RTK-RAS mutations as relapse drivers should be further evaluated with a larger cohort.”

Although only one diagnosis/relapse pair was evaluable in our cohort, Ma and colleagues recently analyzed 20 pairs of diagnostic and relapse samples of pediatric B-ALL (PMID: 25790293), where they showed that relapse founder clones are often subclonal at diagnosis, with a median population frequency of 7%. Of note, they also demonstrated that subclonal RAS pathway mutations with VAFs at diagnosis of ~2% actually expanded and initiated relapse. Therefore, we speculate that the existence of multiple RTK-RAS mutant subclones in IC2 may increase the probability of at least one of these mutant subclones surviving treatment and leading to relapse. We have now added this speculation to the text in the Discussion section (lines 364-373):

“Recently, Ma and colleagues have shown that the relapse founder clone in pediatric B-ALL often originates from a minor subclone at diagnosis, and that subclonal RAS pathway mutations with low VAFs of ~2% can expand and seed relapse³². Therefore, the existence of multiple different RTK-RAS mutant subclones in IC2 may increase the chances that at least one of these mutant subclones will break through treatment and lead to relapse. In fact, our sequencing of a paired relapse sample from a case of IC2 (UT_INF_001) demonstrated that two RTK-RAS mutations at diagnosis had contributed to relapse in this case. However, disappearance of RTK-RAS mutations at relapse has also been reported⁸, indicating that RTK-RAS wild-type subclones can compete and predominate in such cases, necessitating further investigation into the mechanisms of how RTK-RAS mutant subclones contribute to relapse.”

4. The presence of RAS/FLT3 mutations was not associated with EFS/OS in this study, but the presence of multiple RAS clones was associated with a worse EFS and this seems to be more present in the IC2 cluster. IF the authors focus on samples with the higher VAF, for example >25%, is there a EFS/OS difference then?

We thank the reviewer for this suggestion. We have performed a survival analysis based on the presence of RTK-RAS mutations with VAFs >0.25 but the survival rates were not significantly different (Reviewer Fig. 2). From our original results showing an association between the number of RTK-RAS pathway mutations and clinical outcomes, we speculate that the presence of multiple RTK-RAS mutant subclones may confer a probabilistic advantage to leukemia subclones in overcoming treatment. Please see our comments in response to point 3 above for further discussion.

Reviewer Fig. 2

Reviewer Fig. 2. Survival analysis based on the presence of RTK-RAS pathway mutations with VAFs >0.25.

5. The authors perform network analysis for *IRX1*. How does it compare to *IRX2*? Seems both are highly correlated in the RNASeq analysis.

Thank you for raising this point. Our network analysis identified *C5orf38* gene as the only target (regulon) of *IRX2*. Due to the small regulon size (<15 targets), *IRX2* was one of the transcription factors removed before the master regulator inference step. As the reviewer points out, the expression levels of *IRX1* and *IRX2* show a weak but significant positive correlation (Reviewer Fig. 3a). However, the expression of *IRX1* is 3.3-fold higher than *IRX2* (mean TPM in the *IRX* subtype of 125.9 and 37.9; Reviewer Fig. 3b), which presumably led to the difference in the network analysis.

Reviewer Fig. 3

Reviewer Fig. 3. **a**, Correlation analysis of *IRX1* and *IRX2* expression. The regression line is drawn in black and the 95% confidence interval is shaded in gray. Pearson's correlation coefficient R and P -value are indicated. **b**, Expression levels (TPM) of *IRX1* and *IRX2*.

6. Survival analysis of *KMT2A-AFF1* fusions in extended figure 1e is not statistically significant and thus should not be stated as “poorer” in the text (lines 131-133).

We agree with the reviewer and have now rephrased the statement in lines 132-135:

“A trend towards the previously reported poor prognosis of IRX-subtype^{13,20} was also observed among *KMT2A-AFF1* cases, whereas no survival differences were observed when all fusion partners were included (Extended Data Fig. 1e,f).”

7. Infant *KMT2Ar* leukemia often has myeloid co-expression and can undergo lineage switch to myeloid disease at the time to relapse. How are myeloid marker genes expression between the ICs? Would one cluster be more likely to undergo a myeloid switch versus another?

In response to the reviewer’s comment, we have evaluated the myeloid marker enrichment using a list of marker genes obtained from the same single-cell atlas of fetal liver hematopoiesis as used in original Figure 2b (Popescu et al., PMID: 31597962). Although no significant difference was observed, we have added this result as Extended Data Fig. 5b, as we agree with the reviewer that myeloid lineage switch is an important issue in the management of *KMT2A-r* infant ALL. We have also added the marker gene list in Supplementary Table 7 and added the following statement in the text accordingly:

“Although myeloid marker co-expression and lineage switch are commonly observed in the treatment of *KMT2A-r* infant ALL, expression of myeloid lineage signatures did not differ significantly between ICs (Extended Data Fig. 5b).” (Lines: 203-205)

Susceptibility of individual ICs to lineage switch is not evaluable in our current study, since only one lineage switch event was recorded in our discovery cohort (scmc09 in IC4).

8. The authors express *KMT2A-AFF1* in embryonic cells from different stages of development to show that expression in HSC or MPP leads to a IRX1 type of leukemia vs expression in a lymphoid progenitor to a HOX subtype. How would expression of one of the other *KMT2A* fusions compare?

We thank the reviewer for this interesting suggestion. We agree that it would be very interesting to create multiple cellular models using different *KMT2A* fusions and explore phenotypic and mechanistic differences. Although this is certainly an area we would like to explore in a future study, we feel it is beyond the scope of this paper.

9. I am interested in this “Molecular switch” between IRX/HOX subtypes. Perhaps beyond the scope of this paper, would be interesting to see if this switch in subtypes happens with relapse.

Thank you for your positive comment and interesting suggestion. We have examined the same pair of diagnostic and relapse sample used in our response to comment 3 above (UT_INF_001) for the expression pattern of *IRX1* and *HOXA9*, and the “IRX/HOXA subtype switch” was not observed in this case (Reviewer Fig. 4). However, as we consider this result premature to make any statement in our current paper, we would like to investigate this in the future with a larger cohort of relapsed cases.

Reviewer Fig. 4

Reviewer Fig. 4. RT-PCR of *IRX1* and *HOXA9* in the diagnostic (D) and relapse (R) sample of UT_INF_001.

10. There is only one supplementary table included. I could not locate Supplementary Table 2 and later.

We apologize for the inconvenience. We have submitted the revised spreadsheet including all supplementary tables with this letter.

Reviewer #4, expertise in sc-RNAseq and DNA methylation analysis (Remarks to the Author):

In this manuscript, the authors have performed RNA sequencing, methylation array analysis, whole exome and targeted deep sequencing on 84 infants with KMT2A-rearranged leukemia, using multi-omics to define five robust integrative clusters (ICs) in KMT2A-rearranged ALL. What's more, they revealed that the number of RAS pathway mutations predicts prognosis and plays an important way in IC2. They provided a more detailed ALL subtypes, which will be beneficial to genomics-guided individualized therapy, and those multi-omics data could be useful ALL resources database. Taken together, I think this manuscript is probably publishable, but major revision is needed.

Major:

1. The cohort only have ALL samples, but the normal sample should be included too, especially for transcriptome, DNA methylome and single-cell data. Though IRX-subtype and HOXA-subtype were identified by RNA-seq, some features of subtype may be shared by normal samples. Using control cohort helps define clearer ALL subtypes. What's more, When the authors validate this clustering results, they should include normal cohort too.

We thank the reviewer for this suggestion. We have now utilized a published RNA sequencing dataset (GSE122982) and a methylation array dataset (GSE45459) of hematopoietic and B-lineage progenitors to compare with our infant ALL dataset. As the reviewer suggests, clustering with normal samples has improved the single-omics clustering and successfully separated the IC4 infants (please see our new Supplementary Fig. 5), which was not clear in the leukemia-only single-omics clustering (Extended Data Fig. 1 and Supplementary Fig. 2). This clustering has shown that, compared to the other ICs, IC4 leukemia has greater similarity to normal B-cell progenitors in terms of expression and methylation of a set of B-cell developmental genes, which

has likely caused the better separation of IC4 by using normal samples. Our extended methylation cohort also recapitulated this similarity between IC4 and normal B-cell progenitors (new Supplementary Fig. 5c). Accordingly, we have added the following statement to the text (lines 197-200):

“Comparison with published RNA sequencing²⁴ and methylation array²⁵ datasets of normal B-cell progenitors further confirmed the more mature status of IC4 by identifying shared expression and methylation signatures between IC4 and normal progenitors (Supplementary Fig. 5).”

Accession information for the published datasets have also been added to the Methods section (lines: 558-559 and 571-573).

Although the addition of normal samples has improved the single-omics clustering by separating IC4, we have also found that the clustering resolution is still better with the original dual-omics clustering approach. This is presumably because the differences within *KMT2A-r* leukemias are relatively small compared to the large differences between *KMT2A-r* leukemias and normal progenitors, and therefore, the leukemia-only dual-omics approach may be more sensitive in capturing the small but significant heterogeneity within *KMT2A-r* infant ALL.

Finally, we have also performed an integrative single-cell analysis using the fetal liver atlas dataset (Popescu et al., PMID: 31597962). Our original observation of developmental heterogeneity within leukemic blasts has been confirmed by leukemia cell projection and pseudotime inference based on the normal B-cell progenitors (please see new Extended Data Fig. 7b-d). Comparison with normal progenitors has also illustrated lineage markers showing ordered sequential expression as well as aberrant early and/or prolonged expression in leukemia cells (Extended Data Fig. 7e-g). Detailed methods have been included in the Methods section (lines 729-738) and the original Extended Data Fig. 6b-g have been moved to new Supplementary Fig. 10. Accordingly, the following statement has been added to the Results section (lines 297-302):

“Nearest neighbor projection and pseudotime inference based on the fetal liver B-cell progenitors further confirmed the developmental hierarchy within leukemic blasts (Extended Data Fig. 7b-d), where leukemia cells showed sequential expression of developmental markers resembling normal B-cell progenitors (Extended Data Fig. 7e,f), while also exhibiting ectopic early and/or prolonged expression of other developmental regulators (Extended Data Fig. 7g).”

2. In Fig.1a, expression heatmap and methylation heatmap looks alike, whether the methylation probes overlapped with expression heatmap genes? If so, can single omics (transcriptome or DNA methylome) do this clusters?

In response to the reviewer’s comment, we have compared the lists of genes and probes used in Fig. 1a. As shown in Reviewer Fig. 1 below, the overlap was less than 5% for both expression and methylation heatmaps. The 13 overlapping genes included *CPNE8*, *HOXA9*, *IRAK2*, *IRX1*, *KCNH8*, *LGR5*, *PDGFC*, *ROR1*, *SLC6A3*, *TNNI3K*, *WFS1*, *ZFH3*, *ZNF503*. Please also see

comment 3 below, where we have further jointly investigated the overlaps of differential expression and methylation. As for single-omics clustering, neither expression- nor methylation-based clustering identified more than two stable clusters as described in lines 122-125 and 139-142 and in Extended Data Fig. 1 and Supplementary Fig. 2.

Reviewer Fig. 1

Reviewer Fig. 1. Venn diagram of the genes and probes used in Fig. 1a. Methylation probes were annotated with the corresponding genes according to the Illumina's v1.2 annotation.

3. The authors do many omics, such as transcriptome, DNA methylome and WES, but the analysis of each omics is separate, and the authors can try a joint analysis to get more novelty.

We thank the reviewer for raising this important point. We agree that our downstream cluster comparisons were not fully omics-integrative, and we have now performed an integrative analysis to identify dual-omics marker genes that exhibit both significant differential expression and methylation. Indeed, as the reviewer suggests, the integrative analysis has helped to refine the cluster markers, and consequently, *FLT1* gene (encoding VEGFR1) has emerged as a novel dual-omics marker of IC2. It is also important that key hematopoietic marker genes, such as *CD34*, have been confirmed to be not only differentially expressed but also differentially methylated between ICs. These new results have been added as new main Fig. 2 and new Extended Data Fig. 4. The list of dual-omics marker genes has also been added as new Supplementary Table 6. Please see comment 4 below for further discussion.

4. little investigation of the molecular heterogeneity of KMT2A-r infant ALL in DNA methylation. The authors identified more refined subtypes based on both transcriptome and DNA methylation, which indicated that the DNA methylation might be an important determinant of molecular profiles in KMT2A-r infant ALL. Thus, they need to explore the DNA methylation landscape more deeply and in more detail.

We thank the reviewer for this important suggestion. We have now explored the global and local DNA methylation profiles in greater detail. In terms of global methylation status, the five ICs showed similar methylation distributions across the genomic tiling regions as well as gene and promoter loci (please see our new Extended Data Fig. 4a,b). This suggests that small and specific target loci, rather than genome-wide drastic differences, distinguish the methylomes of ICs. Regarding the local methylation differences, thanks to the reviewer's suggestion, the joint transcriptome-methylome comparisons have provided a refined list of dual-omics cluster markers

with both differential expression and methylation, as discussed in comment 3 above. We have further created new Supplementary Fig. 4 to show variable methylation profiles of B-cell marker genes. To describe these new results for this comment and comment 3 above, we have revised the text in lines 170-189:

“Since the addition of methylation information improved the clustering resolution, we next examined the genome-wide DNA methylation status of these five ICs. Globally, DNA methylation levels showed similar distributions between ICs across the genome as well as within gene and promoter loci (Extended Data Fig. 4a,b), suggesting that small sets of IC-specific target loci, rather than genome-wide drastic differences, distinguish the IC methylomes. To define IC-specific markers, we next jointly analyzed the transcriptome and DNA methylome of our discovery cases. First, expression-based marker genes were identified for each cluster (Supplementary Table 5), which were further narrowed down based on differential methylation status. Consequently, 38, 32 and 73 genes were identified as dual-omics marker genes of IC2, IC3 and IC4, respectively (Fig. 2a,b and Extended Data Fig. 4c and Supplementary Table 6), although no genes were identified as dual-omics markers for IC1 and IC5. Among the dual-omics markers, *FLT1* showed the greatest methylation reduction and concordant upregulation in IC2 (Fig. 2c). *FLT1* encodes vascular endothelial growth factor receptor 1 (VEGFR1), which is not physiologically expressed in hematopoietic progenitors²² (Extended Data Fig. 4d) but is reported to be responsible for intra-bone marrow localization and survival of ALL cells²³. Furthermore, key hematopoietic and B-lineage markers, including *CD34*, *MME* (encoding CD10) and *DNMT*, were found among the dual-omics marker genes (Fig. 2d and Extended Data Fig. 4e), suggesting that different ICs have different developmental status towards the B-cell lineage. In fact, conventional B-cell developmental marker genes were variably expressed and methylated between ICs (Supplementary Figs. 3 and 4).”

Minor:

1. For Fig2g, Endothelial signature gene set is small number, and it hard to convincing. It's suggested for authors to collect better Endothelial signature gene set and re-analyze it.

Thank you for making this point. We have now utilized an additional endothelial signature gene set (199 genes) defined by Zeng, et al. in their single-cell transcriptome study of human embryonic hemato-endothelial development (PMID: 31501518). Gene set enrichment was re-analyzed including this new gene set, and the result has been added to Figure 3f. The list of 199 signature genes has also been included in Supplementary Table 7. We have therefore revised the text in lines 214-216:

“Accordingly, endothelial cell signatures derived from human fetal liver²² as well as early human embryonic development²⁷ were enriched in the IRX subtype compared with the HOXA subtype (Fig. 3f)”

2. The authors have showed that enrichment analysis of hypo-methylated regions in HOXA subtype and IRX subtype, how about hyper-methylated regions enrichment analysis?

We apologize for the confusion. Since Figure 2h and 2i (now Figure 3g and 3h) show a comparison between IRX subtype and HOXA subtype, hypo-methylated regions in IRX subtype can be interpreted as hyper-methylated regions in HOXA subtype, and vice versa.

3. The authors successfully identified five subgroups in KMT2A-r infant ALL by both transcriptome and DNA methylome-based clustering, while only two subgroups identified by transcriptome-based clustering. Their results demonstrated that multi-omics analysis performed better than single-omics clustering in revealing more complex and clinically relevant disease subtypes. The reason should be given in the discussion section.

We thank the reviewer for this suggestion to highlight the reasons why multi-omics clustering performs better in subclass discovery. We have added the following statement in the Discussion section (lines: 348-352).

“Since SNF identifies both consistent and complementary patient-to-patient similarities across multiple omics layers²¹, it is most likely that our SNF-based clustering approach successfully integrated the shared and distinct patient similarities captured with transcriptome and methylome information, which were, however, not distinct enough to cluster in a single-omics analysis.”

Reviewers' Comments:

Reviewer #1:

Remarks to the Author:

Thank you for your careful consideration of the critiques and suggestions that were offered. Your responses were appropriate and the revisions that were incorporated into the manuscript have improved the clarity and conclusions obtained.

Reviewer #2:

Remarks to the Author:

The authors have appropriately addressed all my previous questions. No further questions from me.

Reviewer #3:

Remarks to the Author:

The authors addressed all of my comments completely. they also addressed the extensive comments from the other 3 reviewers. This work should be accepted and published.

Reviewer #4:

Remarks to the Author:

The authors have adequately addressed my major concerns. No further comments.

Reviewer #1 (Remarks to the Author):

Thank you for your careful consideration of the critiques and suggestions that were offered. Your responses were appropriate and the revisions that were incorporated into the manuscript have improved the clarity and conclusions obtained.

Thank you very much for your previous suggestions.

Reviewer #2 (Remarks to the Author):

The authors have appropriately addressed all my previous questions. No further questions from me.

Thank you very much for your previous suggestions.

Reviewer #3 (Remarks to the Author):

The authors addressed all of my comments completely. they also addressed the extensive comments from the other 3 reviewers. This work should be accepted and published.

Thank you very much for your previous suggestions.

Reviewer #4 (Remarks to the Author):

The authors have adequately addressed my major concerns. No further comments.

Thank you very much for your previous suggestions.